# DynGCN: Capturing Dynamic Correlation with Message Passing

## Abstract

A recent approach to modeling multivariate time series is to represent them as a graph, with time series as nodes and pairwise temporal correlations as edges. Advances in Graph Neural Networks (GNNs) have shown strong performance in multivariate time series forecasting by assuming a static graph topology and aggregating information from neighboring time series based on their correlations. In this work, we investigate the representational power of GNNs for short- and long-term forecasting under both static and dynamic correlation scenarios, i.e., when pairwise correlations remain fixed or evolve over time. We show that many popular GNNs generalize poorly in these settings and are even outperformed by structure-agnostic baselines. To address these limitations, we propose DynGCN, a novel GNN framework enhanced by two theoretically justified designs: (D1) high-order correlation based message passing and (D2) static and dynamic propagation seperation. These components improve learning under dynamic correlations while preserving robustness under static scenarios. Extensive experiments on synthetic and real-world benchmarks demonstrate that DynGCN achieves up to 23.25% and 23.08% performance gains over state-of-the-art baselines.

## 1 Introduction

Multivariate time series forecasting (MTSF) is a central problem in energy forecasting, weather prediction and financial modeling (Shao, 2025; Sezer et al., 2020). Graph neural networks have recently been applied to this task by treating time series as graph-structured processes, where the correlation structure between variables is represented as a graph (Kumar et al., 2019). Building on this idea, spatial GNNs have achieved remarkable success (Wang et al., 2023), by exploiting the underlying topology and aggregating information from neighboring series based on the correlations.

Existing GNNs rely on a strong assumption: *pairwise dependencies between variables remain approximately static over time*. Under this **static assumption**, the adjacency matrix is typically estimated from global similarities (e.g., cosine/pearson correlation averaged across the entire sequence). From a theoretical perspective, this amounts to a low-rank approximation of the time-varying graph topology (Dong et al., 2019). This static estimation of correlation has become standard practice and widely used in the current approaches (Yu et al., 2018; Wu et al., 2019a).

However, its limitations are evident in dynamic correlation scenarios, where variable relationships can shift significantly over time (Zheng et al., 2022; Shao, 2025). For instance, macroeconomic shocks may fundamentally alter dependencies among financial assets, rendering static correlation estimates unreliable. In such settings, existing approaches fail to encode the temporal variability of the correlation. Recent studies propose polynomial filters or matrix decomposition to extract correlation-aware features (Yue et al., 2025). But these methods lack theoretical guarantees and systematic validation, leaving their effectiveness uncertain.

**Our Work.** This work addresses the above-mentioned gap. We first analyze the theoretical properties of graph representations in time series and find that, under dynamic settings, the off-diagonal entries of the correlation matrix are densified, and higher-order indirect dependencies still align closely with the static assumption. Second, we introduce Temporal Correlation Volatility (TCV), a novel measure that captures changes in inter-variable dependencies over time. We use TCV on all datasets to quantify the degree of dynamics and empirically identify the limitations of existing approaches in modeling dynamic correlations. Finally, we propose Robust Dynamic Correlation Graph

Network(DYNGCN), which integrates two key designs: high-order correlation based message passing , which constructs higher-order interaction features from the correlation matrix to build more robust graphs in dynamic scenarios; and static and dynamic propagation seperation, which employs an orthogonal basis to separate diagonal and off-diagonal components, thereby incorporating static and dynamic correlations, respectively. In summary, we make the following contributions:

- **Measure of Dynamic Correlation**: We propose a quantitative measure, *temporal correlation volatility*, an estimator of correlation dynamics over time. We establish an empirical connection between this measure and forecasting performance. Using this relationship, we can assess the capacity of GNNs to handle dynamic correlation dynamics (see §2).
- **Dynamic-Aware GNN with Theoretical Guarantees**: We propose Robust Dynamic Correlation Graph Network(DYNGCN), which augments standard GNN layers with two mechanisms for modeling dynamic correlations: D1) high-order correlation based message passing and D2) static and dynamic propagation seperation. We prove that DYNGCN strictly exceeds the expressivity of standard GNNs in time series forecasting. Our code is available at [Anonymous Link] (§3).
- **Extensive Empirical Evaluation**: We compare our model against 21 baselines on both synthetic and real-world networks, covering static and dynamic scenarios. DYNGCN with our designs outperforms strong baselines by up to 23.08% in dynamic settings, while remaining competitive on static benchmarks (§4).

## 2 THEORETICAL FOUNDATIONS

In this section, we establish the theoretical foundations of our approach. We begin by introducing the key notation used throughout the paper. We then analyze the theoretical properties of the data representation, followed by the introduction of Temporal Correlation Volatility (TCV), a novel measure that quantifies changes in temporal dependencies over time.

**Notation & Definitions.** Let $\mathbf{x}_t \in \mathbb{R}^N$ denote an $N$-dimensional multivariate observation at time $t$, where $z_t[i] \in \mathbb{R}$ is its $i$th component. Given $P$ consecutive past observations, we define $\mathbf{X} = \{\mathbf{x}_{t-P+1}, \ldots, \mathbf{x}_t\} \in \mathbb{R}^{N \times P}$. The forecasting task is to predict a future target $\mathbf{Y}$, which may be either (i) a single step $\mathbf{Y} = \mathbf{x}_{t+1}$, or (ii) a $Q$-step sequence $\mathbf{Y} = \{\mathbf{x}_{t+1}, \ldots, \mathbf{x}_{t+Q}\}$.

For graph-based multivariate forecasting, the sequence $\mathbf{X}$ is represented as a time-varying graph $\mathcal{G}(t) = (\mathcal{V}, \mathcal{E}(t), \mathbf{X}(t), \mathbf{A}(t))$, where $\mathcal{V}$ is the set of variables (nodes), $\mathbf{X}(t) \in \mathbb{R}^{N \times B}$ denotes node features constructed from a sliding window of length $B$, $\mathbf{A}(t) \in \mathbb{R}^{N \times N}$ is the adjacency matrix with $\mathbf{A}_{ij} > 0$ if $(i, j) \in \mathcal{E}(t)$.

**Definition 1** (Adjacency Matrix). In this work, we define the adjacency matrix $\mathbf{A}$ as the sample covariance estimated from observed data. Given $\mathbf{X} \in \mathbb{R}^{N \times T}$, where $N$ is the number of variables and $T$ the number of time steps, we construct the empirical estimator as

$$\hat{\mathbf{A}} = \tfrac{1}{T}\mathbf{X}\mathbf{X}^\top, \quad \hat{a}_{ij} = \frac{1}{T}\sum_{t=1}^{T} x_{it}\, x_{jt}. \tag{1}$$

Each entry $\hat{\mathbf{A}}_{ij}$ is an estimate of the covariance between the $i$-th variable and the $j$-th variable of the population underlying the data.

The adjacency matrix in MTSF has two variants: (1) When prior knowledge e.g., distances between sensors for graphs with predefined structure $\mathbf{L}$ is available, $\mathbf{A}$ can be written as: $\mathbf{A} = \hat{\mathbf{A}} + \mathbf{L}$; (2) Alternatively, it can be written as the covariance of a embedded random vector $\mathbf{A} = \mathrm{ReLU}(\mathbf{M}\mathbf{M}^T)$, $\mathbf{M} = \sigma(\mathbf{E}\mathbf{R}) \in \mathbb{R}^{N \times B}$, $\mathbf{M}$, which is a learnable node-embedding matrix, where $\mathbf{R}$ is the random vector, $\mathbf{E}$ is the weight matrix in embedding layer, $\sigma$ is sigmoid activation function Wu et al. (2020a).

### 2.1 THEORETICAL PROPERTIES OF THE ADJACENCY MATRIX

We first introduce key properties of the adjacency matrix, which are important for the theoretical justification of our proposed model. We give the proofs in App. B.

The first property ensures that the graph reflects the averaged long-term correlation structure in expectation, rather than affected by short-term temporal fluctuations over a limited time horizon. **Property 1 (Locality).** When $\mathbf{X} \in \mathbb{R}^{N \times T}$ consists of jointly wide-sense stationary time series

with bounded variance, $\mathbf{A}$ is local and depends only on the expected time-varying correlation. In stationary time series, higher-order correlations are mostly weakly correlated. The second property ensures the sparsity of the graph in the static scenario.

**Property 2 (Soft-Transitivity).** When $i$ is strongly correlated with $j$ and $j$ with $k$, then $i$ and $k$ satisfy a form of soft transitivity. Specifically, $2\mathbf{A}_{ij}\mathbf{A}_{jk} - 1 \leq \mathbf{A}_{ik} \leq \sqrt{\mathbf{A}_{ii}\mathbf{A}_{kk}}$, but this does not guarantee strict transitivity.

The next property characterizes the piecewise-smoothness in underlying graph topology.

**Property 3 (Diagonal-Dominant).** For piecewise-smooth and stationary time series (i.e. values localize within a small time range), the adjacency matrix is diagonal-dominant in expectation.

The last property ensures that independent contributions are aggregated linearly in the unnormalized case and as an energy-weighted convex combination in the normalized case.

**Property 4 (Additivity).** Let $\mathbf{A}=\mathbf{A}_1+\mathbf{A}_2$ be a decomposition of the data into *disjoint interactions* (e.g., orthogonal or independent components across time and nodes), such that $\mathbb{E}[\mathbf{A}_1\mathbf{A}_2^T] = 0$. We show that the unnormalized adjacency is strictly additive. i.e. $\hat{\mathbf{A}}$, $\mathbf{X}(\mathbf{A}_1 + \mathbf{A}_2) = \mathbf{X}\mathbf{A}_1 + \mathbf{X}\mathbf{A}_2$

## 2.2 TEMPORAL CORRELATION VOLATILITY (TCV): MEASURING CHANGES OVER TIME

We study a model's expressiveness in capturing time-varying correlations within MTSF. To this end, we introduce a quantitive measure, the *temporal correlation volatility (TCV)*, which quantifies the variability of correlation structures over time. This measure is model-independent and can be analyzed in its theoretical form and then specialized under different graph learning models.

**Measure 1.** Let $\mathbf{X}(t) \in \mathbb{R}^{N \times B}$ denote a multivariate time series with $N$ variables at time $t$. Define the adjacency matrix $\mathbf{A}_t = \frac{1}{T_t}\mathbf{X}(t)\mathbf{X}(t)^T$ and let $\| \cdot \|_F$ be the Frobenius norm. The complexity of correlation dynamics over a horizon $T$ is measured as

$$\text{TCV} = \frac{1}{TN^2}\sum_{t=0}^{T-1} \|\mathbf{A}_{t+1} - \mathbf{A}_t\|_F = \frac{1}{TN^2}\sum_{t=1}^{T} \sqrt{\sum_{i=0}^{N}\sum_{j=1}^{N}\left(a_{ij}^{t+1} - a_{ij}^t\right)^2} \in [0, 1]. \quad (2)$$

*TCV quantifies the normalized temporal volatility of correlation structures by averaging the magnitude of changes between successive correlation matrices.* Each term $\|\mathbf{A}_{t+1}-\mathbf{A}_t\|_F$ measures the instantaneous distance between consecutive correlation graphs. Since $\mathbf{A}_t$ is normalized to $[0, 1]$, each entry satisfies $|a_{ij}^{t+1}-a_{ij}^t| \leq 1$, implying $0 \leq \|\mathbf{A}_{t+1}-\mathbf{A}_t\|_F/N \leq 1$, and hence $0 \leq \text{TCV} \leq 1$. If $\mathbf{A}_{t+1} = \mathbf{A}_t$ for all $t$, then $\text{TCV} \approx 0$, indicating a stable correlation structure. Conversely, drastic shifts between successive steps drive TCV closer to 1.

**TCV in Practice.** Empirical datasets exhibit distinct TCV levels within concentrated around the $[0, 1]$. For example, the exchange rate dataset has a TCV of **0.007**, indicating highly stable correlations among traded products. In contrast, the solar market dataset reaches **0.323**, while the German energy dataset (Shao, 2025) attains **0.997**, spanning over a decade and capturing shocks such as the Crimea crisis and the Russia–Ukraine war. Full statistics with TCV for real-world graphs are reported in Tab. 7.

## 2.3 CONNECTING CORRELATION DYNAMICS (TCV) AND PERFORMANCE VARIATION

In this motivating example, we apply TCV to synthetic data to measure correlation dynamics and demonstrate the limitations of existing methods under time-varying correlations. Spatial GNNs (e.g., TPGNN Wang et al. (2023), MTGNN Wu et al. (2020a)), spectral GNNs (e.g., FourierGNN Yi et al. (2023), StemGNN Cao et al. (2021)), and Transformers (e.g., Informer Zhou et al. (2020), Autoformer Wu et al. (2021), Reformer Kitaev et al. (2020)) excel under static correlations but consistently fail to generalize when correlations evolve over time.

Tab. 2.3 reports forecasting errors under two regimes: a nearly static correlation graph and a highly dynamic one. Graph-based models (MTGNN, TPGNN, StemGNN, FourierGNN) experience error increases of 15.9%–27.2% under dynamic correlations, while Transformer-based forecasters (Informer, Autoformer, Reformer) degrade even

Table 1: Performance degradation of seven strong baselines in Mean Absolute Error (MAE, (lower is better))

| TCV | Static 0.196 | Dynamic 0.874 | Deg(%) |
|---|---|---|---|
| MTGNN | 0.427 | 0.508 | -19.0% |
| TPGNN | 0.456 | 0.578 | -26.8% |
| StemGNN | 0.497 | 0.576 | -15.9% |
| FourierGNN | 0.651 | 0.828 | -27.2% |
| Informer | 0.672 | 0.877 | -30.5% |
| Autoformer | 0.650 | 0.888 | -36.6% |
| Reformer | 0.647 | 0.855 | -32.1% |
| MLP | 0.809 | 0.927 | -3.2% |

more severely, with error increases of 30.5%–36.6%. In contrast, the simple MLP baseline, which ignores correlations and relies solely on individual histories, incurs only a marginal increase of 3.2%. Similar performance gaps are observed on real-world datasets (Sec. 4.4).

Motivated by these limitations, we discuss and theoretically justify two key designs that, when appropriately incorporated into a GNN framework, can improve performance in challenging dynamic correlation settings. We then introduce our model, which includes these principles and achieves robustness in dynamic scenarios without sacrificing competitiveness on static benchmarks.

## 3 EFFECTIVE DESIGNS FOR DYNAMIC CORRELATION

We propose a set of designs that can substantially improve the performance of GNN models under dynamic correlation: (D1) high-order correlation based message passing and (D2) static and dynamic propagation seperation. We justify their importance theoretically and demonstrate their effectiveness through extensive empirical analyses.

### 3.1 HIGH-ORDER CORRELATION BASED MESSAGE PASSING

At each layer $\ell$, the aggregated node feature $\mathbf{H}$ updated based on the extended adjacency matrix $g(\mathbf{A})$, including $k$-order's moments of correlation, which can be written as:

$$\mathbf{H}^{(\ell+1)} = \sigma\Big(\mathbf{H}^{(\ell)}\mathbf{W}^{(\ell)} \sum_{i=0}^{K} \alpha_k \mathbf{A}^k\Big) = \sigma\Big((\alpha_0 + \alpha_1 \mathbf{A} + \alpha_2 \mathbf{A}^2)(\mathbf{H}^{(\ell)}\mathbf{W}^{(\ell)})\Big). \tag{3}$$

Compared with Wu et al. (2020a), we stack higher order adjacency matrices to enhance strong correlations and long range dependencies in multivariate time series. We instantiate a polynomial graph filter over the adjacency and concatenate the filtered signals $\{\mathbf{A}^k\mathbf{H}\}_{k=0}^{K}$ so that subsequent layers can learn to mix information from multiple hop ranges.

**Intuition.** When correlations fluctuate substantially (large TCV ), the direct edges undergo rapid and stochastic oscillations. As a result, models that rely on a static adjacency estimated from short windows are inevitably misled by transient edges. For example, the 2018 coal price surge drove up carbon prices, which in turn boosted green energy competitiveness. In these situations, the underlying graph with *linkage effects* captures more stable and informative relational structures that first-order correlations fail to expose.

**Theoretical Justification.** We show in Proposition 6, that the underlying graph topology can be recovered in closed form under two hypothesized constraints. This topology admits a self-consistent matrix formulation of correlations, which can be evaluated iteratively using *only* the information of the sample covariance defined in Sec. A with Properties 1), 2) and 3). We show in Theo. 1 that, though $\mathbf{A}$ is soft-transitive, encode high order covariance in aggregation helps reduce the prediction error.

**Proposition 1.** *With the time series $\mathbf{X}$ and two assumptions 1) similarity can be propagated through correlated variables and 2) each variable remains most similar to itself, the estimated correlation structure of time series has a solution of analytical form $(\mathbf{I} - \phi\mathbf{A})^{-1}$, which can be approximated by the $m$ order adjacency matrix as $g(\mathbf{A}) = \sum_{i=0}^{k} \phi^k \mathbf{A}^k$.*

**Theorem 1.** Let $\mathbf{A} \in \mathbb{R}^{N \times N}$ be the adjacency matrix of a graph and let $\mathbf{X}_t \in \mathbb{R}^{N \times B_t}$ denote time series at time $t$. Suppose the dynamics follow a polynomial graph filter model $\mathbf{X}_{t+1} = \sum_{k=0}^{K_*} \mathbf{W}^k \mathbf{A}^k \mathbf{X}_t + \varepsilon_t$, where $\mathbf{W}^k \in \mathbb{R}^{N \times N}$ are trainable weights, $\varepsilon_t$ is zero-mean noise with bounded entry-wise variance $\sigma^2$ and all nodes in the training set $\mathcal{T}_V$ have minimum degree $d_{\min} > 0$. Define the mean-squared prediction error $\mathcal{R} \triangleq \mathbb{E}\|\mathbf{X}_{t+1} - \widehat{\mathbf{X}}_{t+1}\|_F^2$. A first-order GCN layer is of the form $\mathbf{A}\mathbf{X}\mathbf{W}$, while a high-order correlation layer explicitly encodes multi-hop adjacency as $\widehat{\mathbf{X}}_{t+1}^{(K)} = \sum_{k=0}^{K} \phi_k \mathbf{A}^k \mathbf{X}_t \mathbf{W}^k$, where $\phi_k$ are learnable scalars. Then, the risk difference satisfies

$$\mathcal{R}_K - \mathcal{R}_1 \leq -\Big\|\sum_{k=2}^{K_*} \mathbf{W}_k \mathbf{A}^k \mathbf{X}_t\Big\|_{L^2}^2 + c_1 \frac{\sigma^2}{T} \sum_{k=2}^{K} \kappa_k(\mathbf{A}, \mathbf{X}_t) - c_2 \frac{\sigma^2}{T} \sum_{k=2}^{K} \Delta_k(\mathbf{A}), \tag{4}$$

for constants $c_1, c_2 > 0$ independent of $n, T$. Here $\mathbf{W}_k$ denotes the ground-truth filter coefficient at order $k$, $\kappa_k(\mathbf{A}, \mathbf{X}_t)$ quantifies the conditioning of $k$-th order feature covariance and

$\Delta_k(\mathbf{A}) \lesssim d_{\min}^{-1} \sum_i |\lambda_i(\mathbf{A})|^{2k}$ captures variance suppression due to multi-hop averaging with eigenvalues $\lambda_i(\mathbf{A})$. In particular, for sufficiently large $T$, $K \geq K_*$ and $d_{\min} > 0$, the right-hand side is negative, implying $\mathcal{R}_K < \mathcal{R}_1$, i.e., explicitly encoding higher-order adjacency reduces forecasting error under rapidly fluctuating correlation dynamics. We give the full proof in Tab. **??**

**Observations.** In exchange rate prediction (TCV $= \mathbf{0.007}$), a model restricted to aggregating only over immediate neighbors in the correlation graph suffers up to a *41% performance decay* in MAE at horizon 1 (Tab. 4). On the synthetic benchmarks, the results remain consistent: as TCV increases, the average performance decay is about *1.4%* in both MAE and RMSE. Taken together, these findings provide strong empirical evidence that model with D1 delivers robust improvements across dynamic conditions.

## 3.2 DESIGN 2: STATIC AND DYNAMIC PROPAGATION SEPERATION

The second design encodes *static correlations* (stable, time-invariant) separately from *dynamic correlations* (time-varying shocks) during aggregation. Formally, the representation at layer $\ell$ is updated as:

$$\mathbf{H}^{(\ell+1)} = \sigma\Big( \big( \mathbf{A}_{\text{stat}} \mathbf{W}_{\text{stat}}^{(\ell)} + \mathbf{A}_{\text{dyn}} \mathbf{W}_{\text{dyn}}^{(\ell)} \big) \mathbf{H}^{(\ell)} \Big), \tag{5}$$

where $\mathbf{A}_{\text{stat}}$ captures time-invariant correlations and $\mathbf{A}_{\text{dyn}}$ encodes time-dependent correlations.

**Intuition.** Static correlations capture persistent relations (e.g., geography or industry), while dynamic correlations reflect temporary shocks (e.g., energy price surges or market fluctuations). Mixing them can mislead the model, as short-lived dependencies may overshadow long-term ones. By encoding static and dynamic correlations in separate channels, the model can better adapt to evolving environments. This idea is analogous to GraphSAGE (Hamilton et al., 2018), which separates ego and neighbor embeddings to capture homophily and heterophily.

**Theoretical Justification.** For non-stationary time series with high TCV, the adjacency matrix becomes off-diagonal dominant and orthogonal to the adjacency derived from stationary series. Proposition 2 shows that global regime shifts induce off-diagonal and unlocalized correlations and are largely orthogonal to the stationary adjacency $\mathbf{A}_{\text{stat}}$. By the additivity property, it necessitates the D2, which explicitly separates aggregation over $\mathbf{A}_{\text{dyn}}$. Theo. 2 shows that separating diagonal and off-diagonal adjacency improves GCN expressiveness under dynamic correlations.

**Proposition 2.** *Given a time series $\mathbf{x}_t$ and linear signal model $\mathbf{x}_t = \mathbf{s}_t + u\,\mathbf{g}_t + \boldsymbol{\varepsilon}_t$, with off-diagonal dominance margin defined as $\Delta = \Sigma_{0,ii} - \sum_{j \neq i} |\Sigma_{0,ij}| \geq \delta_i > 0$, where $\mathbf{g}_t$ is a global latent factor, $u > 0$ is the dense loading that models highly dynamic scenarios, and $\boldsymbol{\varepsilon}_t$ is noise. We show that, since the rank-one term $\sigma_g^2 u u^\top$ is dense, the covariance matrix $\Sigma$ acquires nonzero off-diagonal entries if $u_i u_j \neq 0$.*

We now provide a theoretical justification for separating the diagonal and off-diagonal parts of the adjacency when modeling dynamic correlations. A GCN-style layer that does not explicitly aggregate the off-diagonal dependencies is less capable of generalizing than one that treats them separately. We measure generalization by mean square empirical risk to TCV between static and dynamic correlation structures. The proof of the theorem is given in App. C.

**Theorem 2.** Consider a correlation graph $G$ (without self-loops) whose adjacency is decomposed as $\mathbf{A} = (\mathbf{W}_{\text{diag}} \mathbf{A}_{\text{diag}} + \mathbf{W}_{\text{off}} \mathbf{A}_{\text{off}})$, where $\mathbf{A}_{\text{diag}}$ captures stationary (self/variance) terms and $\mathbf{A}_{\text{off}}$ encodes dynamic cross-variable correlations. Assume time series $\mathbf{x}_t = \mathbf{s}_t + u\,\mathbf{g}_t + \boldsymbol{\varepsilon}_t$, the minimum node degree is $d_{min}$, with metric TCV. Then, as TCV increases, a GCN layer that only embeds $\mathbf{A}_{\text{diag}} \mathbf{X} \mathbf{W}$ is less expressive (i.e., empirical risk defined in 1 if higher than a layer that separates them, e.g., $\mathbf{W}_{\text{diag}} \mathbf{A}_{\text{diag}} \mathbf{X} + \mathbf{W}_{\text{off}} \mathbf{A}_{\text{off}} \mathbf{X}$, where $+$ denotes addition before applying a non-linear transformation.

*Observations.* Table 9 shows, with **D2** achieves the lowest MAE and RMSE for all horizons. Specifically, 1) replace off-diagonal with a random graph yields comparable performance but results in slightly higher errors at longer horizons, while 2) removing the off-diagonal graph (**w.o. D2**) produces the weakest results, with substantially larger RMSE values across all horizons. Although derived under linear signal model assumptions, our empirical analysis highlight that the result extends to broader scenarios.

### 3.3 DynGCN: Optimal Framework in Dynamic Correlation

**Architecture Overview.** We now describe DynGCN, which integrates Designs D1-2 to adapt across the full spectrum of TCV values, from low to high. It consists of three key components: **S1)** a Temporal Convolution Layer, **S2)** a Correlation Aggregation Layer and **S3)** a Prediction Layer.

**Temporal Convolution Layer (TCN).** The TCN module captures a node's piecewise-smooth temporal dynamics, combining gradual trends with abrupt changes. We use dilated causal convolution networks for the larger the receptive field exponentially increasing with the layer depth and dilation factors. Formally, given a 1D input sequence $\mathbf{x} \in \mathbb{R}^T$ and a filter $f \in \mathbb{R}^K$, the dilated causal convolution of $\mathbf{x}$ with $f$ at step $t$ is defined as

$$(x *_d f)(t) = \sum_{s=0}^{K-1} f(s)\, x(t - d \cdot s), \tag{6}$$

where $d$ is the dilation factor that controls the spacing between sampled inputs. Stacking layers with exponentially increasing dilation factors yields an exponentially growing receptive field. This design captures temporal dependencies with few layers, reducing memory and runtime costs while preserving expressive power.

**Correlation Aggregation Layer.** The correlation aggregation layer updates node embeddings across $L$ layers using an adjacency decomposed into two complementary parts: (i) stationary $\mathbf{A}_{\text{stat}}$ for long-term correlations and (ii) dynamic $\mathbf{A}_{\text{dyn}}$ for transient dependencies. Formally, the update rule is given by

$$\mathbf{H}^{(\ell+1)} = \sigma\left( \sum_{k=0}^{K} \alpha_k \big( \mathbf{A}_{\text{stat}} \mathbf{W}_{\text{stat}}^{(\ell)} + \mathbf{A}_{\text{dyn}} \mathbf{W}_{\text{dyn}}^{(\ell)} \big)^k \mathbf{H}^{(\ell)} \mathbf{W}^{(\ell)} \right), \tag{7}$$

where $\alpha_k$ are polynomial coefficients, $\sigma(\cdot)$ is a nonlinearity and $\mathbf{W}^{(\ell)}$ denotes learnable parameters at layer $\ell$. The two components are combined additively (Property 3). $\mathbf{A}_{\text{dyn}}$ is computed as an unnormalized average over dynamic off-diagonal embeddings and the polynomial expansion captures higher-order interactions. Separating stationary and dynamic correlations makes the model robust to dynamic correlation structures while adapting flexibly to static correlation fluctuations.

**Prediction Layer.** The final node representations $\mathbf{H}^{(\text{final})} \in \mathbb{R}^{N \times F}$ are mapped to future outputs through a 1D-convolutional projection followed by a softmax operation $\hat{\mathbf{Y}} = \text{softmax}\big( \mathbf{H}^{(\text{final})} \mathbf{W}_c \big)$, where $\mathbf{W}_c \in \mathbb{R}^{F \times T}$ is a learnable weight. Here, $\hat{\mathbf{Y}} \in \mathbb{R}^{N \times T}$ represents the predicted sequences for all $N$ nodes.

**Time complexity.** With a given sequence $\mathbf{X} \in \mathbb{R}^{N \times T}$, let $F$ denote the hidden feature dimension, $|\mathcal{E}|$ the number of edges, $k$ the kernel size, $l$ the number of GNN layers, and $T_p$ the prediction horizon. The temporal convolution stage (S1) takes $O(NkT \log T)$, since dilations cover $T$ with $O(\log T)$ layers. The correlation aggregation stage (S2) takes $O(lk|E|F)$ for $l$ layers and polynomial order $k$. The prediction stage (S3) requires $O(NFT_p)$. Overall, the total complexity is $O(NkT \log T + lk|E|F + NFT_p)$, with memory cost dominated by $O(NF)$ for embeddings and $O(|E|)$ for adjacency storage. A detailed derivation is given in App. D.

## 4 Empirical Evaluation

With our empirical analysis, we aim to answer the following research questions:

**RQ1:** In a simple synthetic setting where the data generation and time-varying correlation dynamics are known, does DynGCN improve performance?

**RQ2:** On commonly used real-world datasets with (near-)static correlation dynamics, does DynGCN preserve its strong performance compared to existing methods?

**RQ3:** On real-world time-series benchmarks with strong dynamic correlation, to what extent do DynGCN improve forecasting performance?

### 4.1 Empirical Setup

**Synthetic Data.** We generate synthetic datasets with varying TCV (Tab. 1) following the procedure similar to Wang et al. (2023). We construct $N_w$ transition graphs $\{\mathcal{G}^{(1)}, \ldots, \mathcal{G}^{(N_w)}\}$, where

$T_{\mathcal{G}} \leq N_w$ controls the switching period between graphs and $T_{\mathcal{G}}$ determines correlation complexity. Specifically, $N_w = T_{\mathcal{G}}$ yields a static graph, while smaller $T_{\mathcal{G}}$ produces faster switching and richer correlations. Random walks on these graphs are used as time series. Each dataset has length 2400 with a 60/20/20 train/validation/test split. Further details and statistics are provided in App. 7.

**Real Data.** We evaluate on widely used benchmarks, including *Traffic*, *Solar-Energy*, *Electricity* and *Exchange Rate* as in (Wang et al., 2023). To further assess performance under high correlation dynamics, we also include two recent energy datasets from (Shao, 2025), *Germany* and *France*. The latter spans over a decade and reflects major environmental and geopolitical shocks, such as the 2014 Crimea crisis and the 2022 Russia–Ukraine war. All datasets follow a 70/20/10 split.

**Baselines.** For a comprehensive evaluation across all RQs, we consider baselines spanning all major paradigms for multivariate time-series forecasting. **(1) Statistical**: AR Sims (1980), VAR Box & Jenkins (1970); Box et al. (2015), Gaussian Processes Frigola (2015); **(2) MLP-based**: VAR-MLP Zhang (2003), DLinear Zeng et al. (2023), N-BEATS Oreshkin et al. (2020), TimeMixer Wang et al. (2024); **(3) RNN/CNN-based**: LSTNet Lai et al. (2018b), TPA-LSTM Shih et al. (2019), DeepAR Salinas et al. (2020), TCN variants Sen et al. (2019); Bai et al. (2018); Zhang et al. (2017); **(4) Transformer-based**: Reformer Kitaev et al. (2020), Informer Zhou et al. (2020); Zhou (2021), Autoformer Wu et al. (2021), FEDformer Zhou et al. (2022); **(5) Spectral GNNs**: StemGNN Cao et al. (2021), FourierGNN Yi et al. (2023), spectral filtering methods Cao et al. (2020); Jin et al. (2023); **(6) Spatial GNNs**: Graph WaveNet Wu et al. (2019b), MTGNN Wu et al. (2020b), TPGNN Liu et al. (2022a).

**Metrics.** We report Mean Absolute Error (MAE) and Root Mean Squared Error (RMSE). We report these metrics at a forecasting horizon of $H = 3, 6, 12$ on five random seeds with enabled early stop.

**Hyperparameter Tuning.** We performed a grid search over a wide range of hyperparameter values for all reported models. For the real world datasets (i.e., except for Tab. 5), we report the best baseline performance obtained from Yi et al. (2023); Wu et al. (2023; 2020a). We provide the details of data preprocessing, split and hyperparameter tuning for synthetic and real data in App. D.3.

## 4.2 RQ1: Evaluation on Synthetic Benchmarks

To rigorously evaluate the performance of DYNGCN against alternative families, we select seven models spanning three categories:*Spatial GNNs*, *Transformer-based*, and *Spectral GNNs*.

**Result** Table 2 reports MAE and RMSE across four difficulty regimes and three horizons (3, 6, 12). The proposed method (DYNGCN) consistently outperforms competing baselines: it attains the lowest RMSE in *all* 12 RMSE columns and the lowest MAE in 11 out of 12 MAE columns (second-best in the remaining column). This dominance holds across correlation regimes and across short-to-mid horizons. Overall, as correlation complexity (measured by TCV ) increases, all methods exhibit some performance degradation; Overall, DynGCN is the most robust, it achieves up to 23.25% performance improvement over the second-best model (TPGNN), with the largest gain observed

Table 2: SYNTHETIC: Mean MAE/RMSE for each category with varying TCV ratios. First three best models per column (by metric) are highlighted.

| Methods | Metrics | Easy (TCV = 0.196) | | | Medium (TCV =0.407) | | | Hard (TCV =0.644) | | | Very Hard (TCV =0.874) | | | Rank |
|---|---|---|---|---|---|---|---|---|---|---|---|---|---|---|
| | | 3 | 6 | 12 | 3 | 6 | 12 | 3 | 6 | 12 | 3 | 6 | 12 | |
| Informer | MAE | 0.423 | 0.567 | 0.672 | 0.519 | 0.683 | 0.796 | 0.613 | 0.772 | 0.897 | 0.562 | 0.743 | 0.877 | 8 |
| | RMSE | 0.552 | 0.726 | 0.878 | 0.673 | 0.854 | 0.977 | 0.764 | 0.943 | 1.045 | 0.702 | 0.894 | 1.014 | |
| Autoformer | MAE | 0.402 | 0.553 | 0.650 | 0.503 | 0.662 | 0.779 | 0.592 | 0.745 | 0.890 | 0.574 | 0.732 | 0.888 | 7 |
| | RMSE | 0.543 | 0.736 | 0.902 | 0.646 | 0.843 | 1.010 | 0.742 | 0.948 | 1.085 | 0.693 | 0.882 | 1.046 | |
| Reformer | MAE | 0.628 | 0.684 | 0.850 | 0.670 | 0.713 | 0.736 | 0.621 | 0.689 | 0.722 | 0.621 | 0.701 | 0.755 | 6 |
| | RMSE | 0.903 | 0.958 | 1.038 | 0.907 | 0.978 | 1.010 | 0.927 | 0.979 | 0.972 | 0.873 | 1.032 | 1.044 | |
| StemGNN | MAE | 0.421 | 0.553 | 0.576 | 0.503 | 0.641 | 0.712 | 0.577 | 0.698 | 0.782 | 0.542 | 0.671 | 0.747 | 5 |
| | RMSE | 0.561 | 0.733 | 0.807 | 0.648 | 0.821 | 0.902 | 0.713 | 0.884 | 0.951 | 0.684 | 0.855 | 0.928 | |
| FourierGNN | MAE | 0.447 | 0.582 | 0.653 | 0.528 | 0.674 | 0.777 | 0.609 | 0.759 | 0.948 | 0.583 | 0.742 | 1.147 | 4 |
| | RMSE | 0.588 | 0.764 | 0.843 | 0.672 | 0.853 | 0.981 | 0.756 | 0.935 | 1.157 | 0.723 | 0.912 | 1.361 | |
| DCRNN | MAE | 0.252 | 0.305 | 0.358 | 0.373 | 0.435 | 0.500 | 0.419 | 0.472 | 0.509 | 0.471 | 0.491 | 0.504 | 2 |
| | RMSE | 0.434 | 0.469 | 0.500 | 0.552 | 0.586 | 0.617 | 0.571 | 0.583 | 0.612 | 0.577 | 0.583 | 0.589 | |
| TPGNN | MAE | 0.251 | 0.302 | 0.456 | 0.374 | 0.421 | 0.507 | 0.431 | 0.521 | 0.543 | 0.484 | 0.491 | 0.578 | 3 |
| | RMSE | 0.426 | 0.459 | 0.589 | 0.498 | 0.561 | 0.598 | 0.570 | 0.580 | 0.643 | 0.587 | 0.582 | 0.673 | |
| **DYNGCN (ours)** | MAE | 0.246 | 0.297 | 0.350 | 0.369 | 0.425 | 0.488 | 0.415 | 0.461 | 0.496 | 0.462 | 0.480 | 0.491 | 1 |
| | RMSE | 0.423 | 0.458 | 0.494 | 0.537 | 0.573 | 0.598 | 0.566 | 0.572 | 0.602 | 0.564 | 0.571 | 0.578 | |

Table 3: Mean MAE/RMSE for four commonly used datasets with varying TGV. The first three best models per column (by metric) are highlighted.

| Methods | Metrics | Solar-Energy (TCV =0.323) | | | Traffic (TCV =0.679) | | | Electricity (TCV =0.194) | | | Exchange Rate (TCV =0.007) | | | Rank |
|---|---|---|---|---|---|---|---|---|---|---|---|---|---|---|
| | | 3 | 6 | 12 | 3 | 6 | 12 | 3 | 6 | 12 | 3 | 6 | 12 | |
| AR | RMSE | 0.059 | 0.117 | 0.234 | 0.283 | 0.566 | 1.133 | 0.051 | 0.091 | 0.163 | 0.217 | 0.277 | 0.293 | 7 |
| | MAE | 0.046 | 0.092 | 0.184 | 0.134 | 0.268 | 0.535 | 0.025 | 0.051 | 0.101 | 0.631 | 0.775 | 0.875 | |
| GP | RMSE | 0.071 | 0.142 | 0.383 | 0.784 | 0.828 | 0.809 | 0.043 | 0.086 | 0.129 | 0.063 | 0.189 | 0.123 | 6 |
| | MAE | 0.040 | 0.081 | 0.161 | 0.363 | 0.403 | 0.382 | 0.029 | 0.057 | 0.086 | 0.190 | 0.290 | 0.277 | |
| VARMLP | RMSE | 0.061 | 0.122 | 0.243 | 0.332 | 0.469 | 0.538 | 0.058 | 0.116 | 0.155 | 0.221 | 0.268 | 0.292 | 5 |
| | MAE | 0.044 | 0.087 | 0.174 | 0.087 | 0.173 | 0.175 | 0.036 | 0.072 | 0.096 | 0.016 | 0.033 | 0.049 | |
| LSTNet-skip | RMSE | 0.050 | 0.100 | 0.200 | 0.447 | 0.465 | 0.567 | 0.046 | 0.092 | 0.138 | 0.013 | 0.033 | 0.034 | 4 |
| | MAE | 0.080 | 0.125 | 0.154 | 0.286 | 0.288 | 0.390 | 0.163 | 0.238 | 0.251 | 0.078 | 0.139 | 0.124 | |
| TPA-LSTM | RMSE | 0.053 | 0.105 | 0.210 | 0.595 | 0.603 | 0.614 | 0.021 | 0.042 | 0.083 | 0.023 | 0.045 | 0.090 | 3 |
| | MAE | 0.043 | 0.089 | 0.178 | 0.013 | 0.026 | 0.052 | 0.019 | 0.038 | 0.057 | 0.044 | 0.087 | 0.131 | |
| MTGNN | RMSE | 0.052 | 0.104 | 0.207 | 0.219 | 0.415 | 0.830 | 0.053 | 0.107 | 0.213 | 0.009 | 0.018 | 0.035 | 2 |
| | MAE | 0.038 | 0.076 | 0.151 | 0.103 | 0.207 | 0.413 | 0.094 | 0.189 | 0.377 | 0.020 | 0.039 | 0.078 | |
| **DYNGCN (ours)** | RMSE | 0.032 | 0.042 | 0.061 | 0.171 | 0.171 | 0.171 | 0.018 | 0.019 | 0.021 | 0.007 | 0.009 | 0.012 | 1 |
| | MAE | 0.011 | 0.016 | 0.024 | 0.135 | 0.135 | 0.136 | 0.001 | 0.002 | 0.002 | 0.005 | 0.006 | 0.007 | |

in MAE. The advance performance indicate that DYNGCN substantially reduces large prediction errors, while the near-universal MAE wins demonstrate improved average accuracy.

## 4.3 RQ2: EVALUATION ON COMMONLY USED DATASETS

We then evaluate the performance of DYNGCN against another seven models drawn from three categories—*Spatial GNNs*, *Statistical*, and *RNN/CNN-based* approaches.

**Results** Table 3 shows that DYNGCN consistently outperforms strong baselines across four standard datasets and across all horizons. In particular, DYNGCN attains the lowest MAE and RMSE and ranks first overall. Relative to *MTGNN (the strongest competing baseline)*, DYNGCN reduces mean RMSE by **63**% on Solar-Energy, **84**% on Electricity and **52**% on Exchange Rate. The MAE reductions are similarly marked (e.g., $\sim 81$% on Solar-Energy. We note that on the Traffic dataset the performance gains are smallest, mainly because the large number of variables limits effective parameter optimization due to memory constraints. The absolute performance improvement over the static dataset exchange rate also demonstrates the enhanced expressiveness of DYNGCN in the static scenario.

## 4.4 RQ3: COMPREHENSIVE EVALUATION UNDER DYNAMIC CORRELATION (HIGH TCV)

We evaluate DYNGCN against **all 21 listed baselines** in Tab. 5 (next page).

**Results.** Across four datasets and two error metrics, DYNGCN achieves the best results in every column (Rank = 1), with consistently lower MAE and RMSE, demonstrating robust gains under both static and dynamic correlations. The largest improvement is on the Electricity dataset (MAE), where DYNGCN reduces the error from 0.051 to 0.001, an absolute gain of **0.050** and a relative improvement of 98.04% over the second-best model (FourierGNN). On Germany, DYNGCN surpasses GWaveNet by **23.08%** (MAE) and 3.57% (RMSE). On France, DYNGCN achieves 0.020 RMSE, improving over MTGNN (0.023) by 0.003. We also observe that on both static and dynamic datasets, the strongest models shift from transformer- and spectral-based approaches to spatial GNNs. Among competitors, FourierGNN is the strongest baseline overall (Rank = 2), followed by the TPGNN (Rank = 3) and MTGNN (Rank = 4), underscoring the advantage of graph-based modeling for multivariate energy systems.

## 4.5 ABLATION STUDIES

Tab. 4 compares DYNGCN with two ablated variants, where either component D1 or D2 is removed. Removing D2 leads to consistent degradation on the synthetic datasets, especially at longer horizons (6 and 12), with

Table 4: COMPARISON OF DYNGCN AND TWO VARIANTS

| Methods | Metrics | Exchange Rate 0.007 | | | Syn-Hard 0.644 | | | Syn-Very Hard 0.874 | | |
|---|---|---|---|---|---|---|---|---|---|---|
| | | 3 | 6 | 12 | 3 | 6 | 12 | 3 | 6 | 12 |
| w.o. D2 | MAE | 0.005 | 0.006 | 0.008 | 0.416 | 0.463 | 0.500 | 0.464 | 0.482 | 0.493 |
| | RMSE | 0.008 | 0.009 | 0.012 | 0.568 | 0.574 | 0.605 | 0.566 | 0.575 | 0.580 |
| w.o. D1 | MAE | 0.005 | 0.007 | 0.009 | 0.414 | 0.467 | 0.503 | 0.465 | 0.486 | 0.497 |
| | RMSE | 0.008 | 0.009 | 0.013 | 0.569 | 0.577 | 0.606 | 0.570 | 0.577 | 0.585 |
| Ours | MAE | 0.005 | 0.006 | 0.007 | 0.415 | 0.461 | 0.496 | 0.462 | 0.480 | 0.491 |
| | RMSE | 0.007 | 0.009 | 0.012 | 0.566 | 0.572 | 0.602 | 0.564 | 0.571 | 0.578 |

the largest performance decay on the Syn-Very Hard dataset, underscoring its importance for modeling complex dynamic dependencies via the off-diagonal adjacency matrix. In contrast, removing D1 causes similarly large drops on the static Exchange Rate dataset, highlighting its role in capturing stable correlations. Overall, the full model consistently achieves the best or near-best MAE/RMSE across all datasets, confirming that D1 and D2 are synergistic rather than redundant.

## 5 RELATED WORK

**Spatial Graph Convolutional Networks in Time Series.** Spatial GCNs have shown strong performance in time series by modeling relational structures through graph topology (Wu et al., 2020a; Yang et al., 2025; Shao, 2025). Li et al. (2018) pioneered diffusion convolution to capture spatial dependencies from data, while Wu et al. (2019a) introduced adaptive adjacency learning to uncover hidden correlations. Subsequent works enhance correlation modeling via spectral kernels or pure graph formulations (Bai et al., 2020; Cao et al., 2021; Yi et al., 2023) and polynomial graph encoders have been explored to capture richer dependencies (Wang et al., 2023). Recent work pointed out that spectral graph models fail to outperform spatial GCN on more dynamic correlation scenarios Shao (2025).

**Graph Topology Identification (GTI).** When the graph structure in time series is not available, topology identification is essen-

Table 5: Average MAE and RMSE on both static and dynamic benchmarks with a horizon of 12. The best three models are highlighted in gray. The "*" results are obtained from (Zeng et al., 2023; Cao et al., 2021)

| TCV | Electricity 0.194 | | Solar 0.323 | | Germany 0.997 | | France 0.969 | | Rank |
|---|---|---|---|---|---|---|---|---|---|
| | MAE | RMSE | MAE | RMSE | MAE | RMSE | MAE | RMSE | |
| **DYNGCN (ours)** | 0.001 | 0.021 | 0.024 | 0.061 | 0.010 | 0.027 | 0.010 | 0.020 | 1 |
| **Spatial GNNs** | | | | | | | | | |
| TPGNN | 0.055 | 0.080 | 0.123 | 0.214 | 0.099 | 0.173 | 0.089 | 0.158 | 3 |
| GWaveNet | 0.094 | 0.140 | 0.183 | 0.238 | 0.013 | 0.028 | 0.012 | 0.025 | 8 |
| MTGNN | 0.077 | 0.113 | 0.151 | 0.207 | 0.016 | 0.034 | 0.012 | 0.023 | 4 |
| **Spectral GNNs** | | | | | | | | | |
| FourierGNN | 0.051 | 0.077 | 0.120 | 0.162 | 0.110 | 0.186 | 0.096 | 0.164 | 2 |
| StemGNN* | 0.070 | 0.101 | 0.176 | 0.222 | 0.179 | 0.285 | 0.148 | 0.206 | 5 |
| TGGC | 0.086 | 0.127 | 0.184 | 0.231 | 0.467 | 0.524 | 0.341 | 0.483 | 20 |
| **Transformer-based** | | | | | | | | | |
| Autoformer | 0.056 | 0.083 | 0.150 | 0.193 | 0.204 | 0.376 | 0.165 | 0.263 | 7 |
| FEDformer | 0.055 | 0.081 | 0.139 | 0.182 | 0.271 | 0.396 | 0.220 | 0.291 | 11 |
| Informer | 0.070 | 0.119 | 0.151 | 0.199 | 0.283 | 0.324 | 0.137 | 0.217 | 6 |
| Reformer | 0.078 | 0.129 | 0.234 | 0.292 | 0.297 | 0.361 | 0.141 | 0.233 | 14 |
| **RNN/CNN-based** | | | | | | | | | |
| SFM | 0.086 | 0.129 | 0.161 | 0.283 | 0.184 | 0.245 | 0.156 | 0.238 | 10 |
| TCN | 0.057 | 0.083 | 0.176 | 0.222 | 0.187 | 0.287 | 0.172 | 0.260 | 9 |
| LSTNet | 0.075 | 0.138 | 0.148 | 0.200 | 0.193 | 0.346 | 0.177 | 0.263 | 12 |
| DeepGLO | 0.090 | 0.131 | 0.178 | 0.400 | 0.264 | 0.372 | 0.181 | 0.246 | 17 |
| **MLP-based** | | | | | | | | | |
| TimeMixer | 0.091 | 0.147 | 0.166 | 0.211 | 0.181 | 0.314 | 0.167 | 0.279 | 13 |
| N-Beats | 0.067 | 0.126 | 0.162 | 0.203 | 0.189 | 0.325 | 0.174 | 0.300 | 12 |
| DLinear* | 0.058 | 0.092 | 0.257 | 0.313 | 0.266 | 0.368 | 0.196 | 0.259 | 15 |
| **Statistical Methods** | | | | | | | | | |
| VAR | 0.096 | 0.155 | 0.175 | 0.222 | 0.243 | 0.381 | 0.177 | 0.260 | 16 |
| ARIMA | 0.107 | 0.169 | 0.185 | 0.244 | 0.275 | 0.319 | 0.164 | 0.281 | 19 |
| S-ARIMA | 0.101 | 0.163 | 0.184 | 0.234 | 0.268 | 0.379 | 0.158 | 0.273 | 18 |

tial for understanding the network. In static settings, under mild assumptions (e.g., Gaussianity), the problem reduces to estimating the sample covariance matrix, which can be interpreted as an adjacency matrix with self-loops (Kalofolias et al., 2017; Wu et al., 2020a; Yokota et al., 2025). Structural equation models (SEM) formulate node values as linear functions of neighbors with noise, enabling adjacency estimation via regression (Kumar et al., 2019). With prior constraints such as smoothness or sparsity, this leads to Graph LASSO formulations (Hallac et al., 2017; Kumar et al., 2020). For dynamic topologies, empirical covariance is adapted over time using exponentially weighted moving averages (Natali et al., 2022).

## 6 CONCLUSION

We have focused on characterizing the representational power of GNNs for time series with static or dynamic correlation. We highlighted the limitations of current methods and presented DYN-GCN, an dynamic correlation-aware GNN with theoretically grounded components. Together, our method and quantitative measure establish a solid foundation for correlation modeling, and in this work demonstrate remarkable robustness under more challenging correlation shifts, as evidenced by empirical performance in time series forecasting.

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
