## A  NOTATION

Table 6: Summary of notation used throughout the paper.

| Symbol | Description |
|---|---|
| $\mathbf{x}_t$ | Multivariate observation at time $t$, $\mathbf{x}_t \in \mathbb{R}^N$ |
| $z_t[i]$ | The $i^{\text{th}}$ variable in $\mathbf{z}_t$, $z_t[i] \in \mathbb{R}$ |
| $N$ | Number of variables (dimension of the multivariate time series) |
| $\mathbf{X}$ | Input sequence of past $P$ observations, $\{\mathbf{z}_{t_1}, \ldots, \mathbf{z}_{t_P}\}$ |
| $P$ | Number of historical time steps used for prediction |
| $\mathbf{Y}$ | Target future value(s), $\{\mathbf{z}_{t_{P+Q}}\}$ or $\{\mathbf{z}_{t_{P+1}}, \ldots, \mathbf{z}_{t_{P+Q}}\}$ |
| $Q$ | Forecast horizon (number of time steps to predict) |
| $\mathbf{S}_{t_i}$ | Feature matrix at time $t_i$, $\mathbf{S}_{t_i} \in \mathbb{R}^{N \times D}$ |
| $D$ | Number of features per variable (1 target + $D-1$ auxiliary features) |
| $\mathcal{X}$ | Input sequence with auxiliary features, $\{\mathbf{S}_{t_1}, \ldots, \mathbf{S}_{t_P}\}$ |
| $f(\cdot)$ | Prediction function mapping $\mathcal{X}$ to $\mathbf{Y}$ |
| $\mathbf{A}$ | Adjacency matrix of the correlation graph |
| $\mathbf{A}_{\text{diag}}$ | Stationary/self-term component of adjacency (variance structure) |
| $\mathbf{A}_{\text{off}}$ | Dynamic cross-variable correlation component of adjacency |
| $\mathbf{W}_{\text{diag}}, \mathbf{W}_{\text{off}}$ | Trainable weights for stationary and dynamic adjacency parts |
| $d_{\min}$ | Minimum node degree in the training set |
| TCV | Dynamic correlation strength (e.g., $\|\mathbf{A}_{\text{off}}\|_F^2$ or $\text{Tr}(H\Sigma_{\text{off}})$) |
| $\sigma^2$ | Upper bound on entry-wise noise variance |
| $u$ | Scalar controlling global/dynamic correlation intensity |
| $\mathbf{s}_t$ | Stationary component of $\mathbf{x}_t$ |
| $\mathbf{g}_t$ | Dynamic/global-shift component of $\mathbf{x}_t$ |
| $\varepsilon_t$ | Zero-mean idiosyncratic noise component of $\mathbf{x}_t$ |
| $\mathcal{R}$ | Mean-squared prediction risk, $\mathbb{E}\|\mathbf{X}_{t+1} - \widehat{\mathbf{X}}_{t+1}\|_F^2$ |
| $H$ | Hessian of a graph-level task $y(\mathbf{X})$ (captures pairwise sensitivities) |
| $\Sigma_{\text{off}}$ | Off-diagonal covariance matrix of $\mathbf{x}_t$ (dynamic correlations) |

## B  THEORETICAL FOUNDATIONS: PROOFS

**Theorem 3** (Locality under Stationarity). *Let $\{X_i(t)\}_{t=1}^T$ for $i = 1, \ldots, d$ be jointly wide-sense stationary, zero-mean, ergodic time series with finite variances*

$$\sigma_i^2 = \mathbb{E}[X_i(t)^2].$$

*Define*

$$A_{ij} = \frac{\sum_{t=1}^T X_i(t)\, X_j(t)}{\sum_{k=1}^d \sum_{t=1}^T X_k(t)^2}.$$

*Then, as $T \to \infty$,*

$$A_{ij} \xrightarrow{\text{a.s.}} \frac{\gamma_{ij}(0)}{\sum_{k=1}^d \sigma_k^2}, \quad \text{where } \gamma_{ij}(0) = \mathbb{E}[X_i(t)X_j(t)].$$

*Hence $A_{ij}$ depends only on the pair $(i, j)$ via their zero-lag covariance, scaled by a constant independent of $(i, j)$: it is local up to a global constant. If, in addition, all $\sigma_i^2 = \sigma^2$ (homoscedastic), then*

$$A_{ij} \xrightarrow{\text{a.s.}} \frac{\gamma_{ij}(0)}{d\,\sigma^2} = \frac{\rho_{ij}}{d},$$

*where $\rho_{ij}$ is the Pearson correlation.*

*Proof.* Write $A_{ij}$ as a ratio of empirical moments:

$$A_{ij} = \frac{\frac{1}{T}\sum_{t=1}^T X_i(t)X_j(t)}{\frac{1}{dT}\sum_{k=1}^d \sum_{t=1}^T X_k(t)^2} \cdot \frac{1}{d}.$$

By stationarity and ergodicity,

$$\frac{1}{T}\sum_{t=1}^{T} X_i(t)X_j(t) \xrightarrow{\text{a.s.}} \gamma_{ij}(0), \qquad \frac{1}{T}\sum_{t=1}^{T} X_k(t)^2 \xrightarrow{\text{a.s.}} \sigma_k^2.$$

$$\frac{1}{dT}\sum_{k,t} X_k(t)^2 = \frac{1}{d}\sum_{k=1}^{d}\left(\frac{1}{T}\sum_t X_k(t)^2\right) \xrightarrow{\text{a.s.}} \frac{1}{d}\sum_{k=1}^{d}\sigma_k^2.$$

Therefore,

$$A_{ij} \xrightarrow{\text{a.s.}} \frac{\gamma_{ij}(0)}{\sum_{k=1}^{d}\sigma_k^2}.$$

The limit depends on $(i,j)$ only through $\gamma_{ij}(0)$; the denominator is constant across all pairs. Hence $A_{ij}$ is pairwise-local up to a global rescaling.

**Step 6 (Role of smoothness).** If each $X_i$ is smooth in time (e.g., low-pass, bounded variation, or strongly mixing), then deviations from the limit shrink faster:

$$\left|\frac{1}{T}\sum_t X_i(t)X_j(t) - \gamma_{ij}(0)\right| = O_{\mathbb{P}}(T^{-1/2}), \qquad \left|\frac{1}{T}\sum_t X_k(t)^2 - \sigma_k^2\right| = O_{\mathbb{P}}(T^{-1/2}),$$

with smaller constants for smoother processes. Thus locality holds approximately at finite $T$ and exactly as $T \to \infty$. $\qquad\square$

**Theorem 4** (Soft transitivity). Let $A \in \mathbb{R}^{d\times d}$ be a symmetric positive semidefinite (PSD) similarity/adjacency matrix. Then for any triple $(i,j,k)$ the following hold:

(i) **Cauchy–Schwarz-type bounds (PSD consistency).**
$$A_{ij}^2 \leq A_{ii}A_{jj}, \qquad A_{ik}^2 \leq A_{ii}A_{kk}, \qquad A_{jk}^2 \leq A_{jj}A_{kk}.$$
Consequently, no pairwise similarity can be arbitrarily large relative to the corresponding self-similarities.

(ii) **Soft transitivity under cosine normalization.** If, in addition, $A$ is a correlation/cosine matrix (i.e., $A_{ii} = 1$ for all $i$ so that $A_{ij} = \cos\theta_{ij}$ for some angles $\theta_{ij} \in [0,\pi]$), then
$$A_{ik} \geq \cos(\theta_{ij} + \theta_{jk}) \geq 2A_{ij}A_{jk} - 1.$$
Thus large $A_{ij}$ and $A_{jk}$ force a quantitative lower bound on $A_{ik}$.

This constitutes *soft transitivity*: triplewise similarities must be mutually compatible but need not satisfy strict threshold-based transitivity.

*Proof.* (i) Since $A$ is PSD, there exist vectors $\{z_\ell\}_{\ell=1}^{d}$ in an inner-product space such that $A_{pq} = \langle z_p, z_q\rangle$ for all $p,q$. Applying the Cauchy–Schwarz inequality to each pair yields $A_{ij}^2 = \langle z_i, z_j\rangle^2 \leq \|z_i\|^2\|z_j\|^2 = A_{ii}A_{jj}$, and analogously for $(i,k)$ and $(j,k)$.

(ii) If $A$ is correlation/cosine normalized, write $u_\ell := z_\ell/\|z_\ell\|$ so $\|u_\ell\| = 1$ and $A_{pq} = \langle u_p, u_q\rangle = \cos\theta_{pq}$ with $\theta_{pq}$ the spherical (angular) distance. On the unit sphere the triangle inequality holds: $\theta_{ik} \leq \theta_{ij} + \theta_{jk}$. Since $\cos$ is decreasing on $[0,\pi]$,
$$A_{ik} = \cos\theta_{ik} \geq \cos(\theta_{ij} + \theta_{jk}).$$
Using the identity $\cos(\alpha + \beta) = \cos\alpha\cos\beta - \sin\alpha\sin\beta$ and $\sin\alpha\sin\beta \leq \sqrt{(1 - \cos^2\alpha)(1 - \cos^2\beta)} \leq 1 - \cos\alpha\cos\beta$, one obtains the looser but simple bound $\cos(\alpha + \beta) \geq 2\cos\alpha\cos\beta - 1$, hence $A_{ik} \geq 2A_{ij}A_{jk} - 1$. $\qquad\square$

**Soft & Strict Transitivity** The inequalities in (i) constrain magnitudes pairwise but do not impose a threshold-type rule such as $A_{ij} \geq \tau \wedge A_{jk} \geq \tau \Rightarrow A_{ik} \geq \tau$. Even under cosine normalization (ii), the best universal lower bound is $A_{ik} \geq 2A_{ij}A_{jk} - 1$, which allows $A_{ik}$ to be small or even negative unless $A_{ij}$ and $A_{jk}$ are sufficiently large. Therefore PSD enforces *indirect consistency* (coherence among triples) but not *strict logical transitivity*.

**Soft Transitivity** Take three unit vectors in $\mathbb{R}^2$ at angles $0°, 60°, 120°$: $u_i = (1, 0)$, $u_j = (\frac{1}{2}, \frac{\sqrt{3}}{2})$, $u_k = (-\frac{1}{2}, \frac{\sqrt{3}}{2})$. Then $A_{ij} = A_{jk} = \cos 60° = \frac{1}{2}$, but $A_{ik} = \cos 120° = -\frac{1}{2}$. Thus for threshold $\tau = \frac{1}{2}$, we have $A_{ij} \geq \tau$ and $A_{jk} \geq \tau$ yet $A_{ik} < \tau$.

**Proposition 3** (Strong Diagonality). *Let* $\mathbf{X} \in \mathbb{R}^{N \times T}$ *collect* $N$ *continuous-valued time series as rows* $x_i^\top$, *and define the normalized Gram adjacency*

$$\mathbf{A} = \mathbf{X}\mathbf{X}^\top$$

*Then:*

(a) ***Diagonal anchoring.*** $\sum_{i=1}^{N} A_{ii} = 1$.

(b) ***Off-diagonals shrink with frequent switching.*** *Assume the time series are generated by a piecewise-stationary, graph-switching process: there exists a regime process* $(S_t)_{t=1}^T$ *taking values in a finite set* $\mathcal{R}$ *(transition graph), such that*

$$x_i(t) = s_i^{(S_t)}(t) + \varepsilon_i(t),$$

*where for each regime* $r \in \mathcal{R}$ *the vector process* $\{s^{(r)}(t)\}_t$ *is zero-mean, WSS within* $r$, *and different regimes are uncorrelated (orthogonal) across time; noises* $\varepsilon_i$ *are zero-mean, finite-variance, and independent across* $i$ *and from* $S_t$. *Let* $\pi_r$ *be the stationary frequency of regime* $r$, *and let* $M$ *be the number of switches of* $S_t$ *over horizon* $T$. *If switching is frequent (i.e., the average segment length* $T/M$ *is bounded and the regime chain is mixing), then for* $i \neq j$,

$$\mathbb{E}[A_{ij}] = \frac{\sum_{r \in \mathcal{R}} \pi_r \gamma_{ij}^{(r)}(0)}{\sum_{k=1}^{N} \sum_{r \in \mathcal{R}} \pi_r \sigma_k^{(r)2} + \sum_{k=1}^{N} \sigma_{\varepsilon,k}^2} \quad \text{and} \quad A_{ij} = O_{\mathbb{P}}\left(\sqrt{\frac{1}{T_{\text{eff}}}}\right),$$

*where* $\gamma_{ij}^{(r)}(0) = \mathbb{E}[s_i^{(r)}(t)s_j^{(r)}(t)]$ *and* $\sigma_k^{(r)2} = \mathbb{E}[s_k^{(r)}(t)^2]$, *and* $T_{\text{eff}}$ *is the effective sample size that* increases *as the switching rate increases (shorter correlation length). In particular, if the cross-correlations average out across regimes (e.g.,* $\sum_r \pi_r \gamma_{ij}^{(r)}(0) = 0$*), then* $\mathbb{E}[A_{ij}] = 0$ *and*

$A_{ij} \xrightarrow{\mathbb{P}} 0$ *faster as switching becomes more frequent, while the diagonal mass remains anchored by (a). Hence* $\mathbf{A}$ *becomes increasingly diagonal-dominant.*

*Proof.* (a) Immediate from the definition:

$$\sum_{i=1}^{N} A_{ii} = \frac{\sum_{i=1}^{N} \|x_i\|^2}{\sum_{k=1}^{N} \|x_k\|^2} = 1.$$

(b) Decompose the time horizon into consecutive segments $\{I_m\}_{m=1}^M$ on which the regime is constant ($S_t = r_m$ for $t \in I_m$). Then

$$\langle x_i, x_j \rangle = \sum_{m=1}^{M} \sum_{t \in I_m} x_i(t)x_j(t)$$

$$= \underbrace{\sum_{m=1}^{M} \sum_{t \in I_m} s_i^{(r_m)}(t)s_j^{(r_m)}(t)}_{\text{within-regime}} + \underbrace{\sum_{m=1}^{M} \sum_{t \in I_m} \left(s_i^{(r_m)}(t)\varepsilon_j(t) + \varepsilon_i(t)s_j^{(r_m)}(t)\right)}_{\text{zero-mean}}$$

$$+ \underbrace{\sum_{t=1}^{T} \varepsilon_i(t)\varepsilon_j(t)}_{\substack{\text{zero-mean} \\ i \neq j}}.$$

By the orthogonality/mixing assumptions, the cross terms are mean zero and concentrate at the $O_{\mathbb{P}}(\sqrt{T_{\text{eff}}})$ scale. For the within-regime term,

$$\mathbb{E}\left[\langle x_i, x_j \rangle\right] = \sum_{r \in \mathcal{R}} \mathbb{E}\left[\sum_{t:S_t=r} s_i^{(r)}(t)s_j^{(r)}(t)\right] = \sum_{r \in \mathcal{R}} \pi_r T \gamma_{ij}^{(r)}(0),$$

while

$$\mathbb{E}\big[\|\mathbf{X}\|_F^2\big] = \sum_{k=1}^{N} \sum_{r \in \mathcal{R}} \pi_r \, T \, \sigma_k^{(r)2} \; + \; \sum_{k=1}^{N} T \, \sigma_{\varepsilon,k}^2.$$

Dividing yields the stated $\mathbb{E}[A_{ij}]$. Frequent switching reduces the correlation length, so $T_{\text{eff}}$ (effective independent samples) increases, sharpening concentration of $A_{ij}$ around its mean. If regime-averaged cross-covariances cancel (e.g., oppositely signed across regimes or weak overall), then $\mathbb{E}[A_{ij}] = 0$ and $A_{ij} \to 0$ in probability as switching intensifies. The denominator is a sum of (non-negative) variances across nodes and regimes, so the diagonal share $\sum_i A_{ii} = 1$ remains fixed by (a); hence relative off-diagonal weight diminishes, yielding diagonal dominance. $\square$

**Remark 1** (When off-diagonals persist). *If regimes share a common correlation backbone so that $\sum_r \pi_r \, \gamma_{ij}^{(r)}(0) > 0$ on a subset of pairs $(i, j)$, those entries remain non-negligible. The matrix is still diagonally anchored ($\text{trace}(A) = 1$), but exhibits structured off-diagonal blocks along the shared backbone.*

**Lemma 1** (Population covariance under global regime shifts). *Under the global-regime model above, the population covariance decomposes as in equation* **??**. *In particular, if $u$ has at least two nonzeros, then $\Sigma$ has nonzero off-diagonal entries wherever $(u_i u_j) \neq 0$.*

*Proof.* Since $s_t$, $g_t$, $\varepsilon_t$ are mutually uncorrelated and $\mathbb{E}g_t = 0$,

$$\Sigma \; = \; \text{Cov}(s_t) + \text{Cov}(ug_t) + \text{Cov}(\varepsilon_t) \; = \; \Sigma_s + \sigma_g^2 uu^\top + \Sigma_\varepsilon.$$

If $u_i \neq 0$ and $u_j \neq 0$ with $i \neq j$, then $(\sigma_g^2 uu^\top)_{ij} = \sigma_g^2 u_i u_j \neq 0$, hence $\Sigma_{ij} \neq 0$ unless exactly canceled by $(\Sigma_s)_{ij}$, which is non-generic if $\Sigma_s$ is banded/diagonal-dominant. $\square$

**Proposition 4** (Off-diagonal densification). *If $u$ has at least two nonzeros, then the rank-one term $\sigma_g^2 uu^\top$ is dense on the support of $u$, hence $\Sigma$ in equation* **??** *exhibits widespread off-diagonal entries. Quantitatively, for any index $i$ with $u_i \neq 0$,*

$$\sum_{j \neq i} \mathbf{1}\{\Sigma_{ij} \neq 0\} \; \geq \; \sum_{j \neq i} \mathbf{1}\{u_j \neq 0\} \; - \; \#\{\textit{coincidental cancellations by } (\Sigma_s)_{ij}\},$$

*which is maximal (dense) when $\Sigma_s$ is diagonal/banded and $u$ is dense.*

*Proof.* Immediate from Lemma 1, since $(\sigma_g^2 uu^\top)_{ij} = \sigma_g^2 u_i u_j$ for $i \neq j$. Cancellations require fine-tuned signs/magnitudes of $(\Sigma_s)_{ij}$ and are non-generic. $\square$

**Proposition 5** (Loss of diagonal dominance). *Suppose $\Sigma_s + \Sigma_\varepsilon$ is diagonally dominant with margins $\Delta_i(\Sigma_s + \Sigma_\varepsilon) \geq \delta_i > 0$, where $\Delta_i(M) = |M_{ii}| - \sum_{j \neq i} |M_{ij}|$. Then for any $u$,*

$$\Delta_i(\Sigma_s + \sigma_g^2 uu^\top + \Sigma_\varepsilon) \; \leq \; \Delta_i(\Sigma_s + \Sigma_\varepsilon) \; + \; \Delta_i(\sigma_g^2 uu^\top).$$

*Moreover, if*

$$\|\sigma_g^2 uu^\top\|_{\infty \to \infty} \; > \; \min_i \delta_i,$$

*then $\Sigma$ is not diagonally dominant.*

*Proof.* Triangle inequality row-wise gives the first inequality. For the second claim, note that $\|M\|_{\infty \to \infty} = \max_i \sum_j |M_{ij}|$ upper-bounds the worst-case row perturbation of diagonal dominance. If the perturbation exceeds the smallest margin $\min_i \delta_i$, at least one row loses diagonal dominance. $\square$

## C    DYNGCN: DETAILS OF THEOREMS

**Proposition 6.** *With the time series $\mathbf{X}$ and two assumptions 1) similarity can be propagated through correlated variables and 2) each variable remains most similar to itself, the estimated correlation structure of time series has a solution of analytical form $(\mathbf{I} - \phi\mathbf{A})^{-1}$, which can be approximated by the $m$ order adjacency matrix as $g(\mathbf{A}) = \sum_{i=0}^{k} \phi^k \mathbf{A}^k$.*

The correlation structure of time series can be expressed recursively under two natural conditions: (1) similarity propagates through correlated variables, and (2) each variable remains most similar to itself. Formally, for adjacency matrix $\mathbf{A}$ derived from pairwise correlations and balancing parameter $\phi \geq 0$, the similarity matrix $\mathbf{S}$ satisfies

$$\mathbf{S} = \phi \mathbf{A}\mathbf{S} + \mathbf{I}.$$

The unique closed-form solution is

$$\mathbf{S} = (\mathbf{I} - \phi\mathbf{A})^{-1} = \mathbf{I} + \phi\mathbf{A} + \phi^2\mathbf{A}^2 + \phi^3\mathbf{A}^3 + \cdots.$$

**Remark 2.** *A general correlation filter can be expressed as a polynomial in $\mathbf{A}$,*

$$g(\mathbf{A}) = \alpha_0\mathbf{I} + \alpha_1\mathbf{A} + \alpha_2\mathbf{A}^2,$$

*which combines self-similarity, direct similarity, and second-order transitive similarity. Higher-order filters naturally extend this form.*

**Theorem 5.** Let $\mathbf{A} \in \mathbb{R}^{n \times n}$ be the adjacency matrix of a graph (without self-loops), and let $\mathbf{X}_t \in \mathbb{R}^{n \times p}$ denote node features at time $t$. Suppose the dynamics follow a polynomial graph filter model

$$\mathbf{X}_{t+1} = \sum_{k=0}^{K_*} \mathbf{W}^k \mathbf{A}^k \mathbf{X}_t + \varepsilon_t,$$

where $\mathbf{W}^k \in \mathbb{R}^{p \times p}$ are trainable weights, $\varepsilon_t$ is zero-mean noise with bounded entry-wise variance $\sigma^2$, and all nodes in the training set $\mathcal{T}_V$ have minimum degree $d_{\min} > 0$.

Define the mean-squared prediction risk

$$\mathcal{R} \triangleq \mathbb{E}\big\|\mathbf{X}_{t+1} - \widehat{\mathbf{X}}_{t+1}\big\|_F^2.$$

- A *first-order GCN layer* is of the form

$$\widehat{\mathbf{X}}_{t+1}^{(1)} = \mathbf{A}\mathbf{X}_t\mathbf{W}.$$

- A *high-order correlation layer* explicitly encodes multi-hop adjacency as

$$\widehat{\mathbf{X}}_{t+1}^{(K)} = \sum_{k=1}^{K} \phi_k \mathbf{A}^k \mathbf{X}_t \mathbf{W}^k,$$

where $\phi_k$ are learnable scalar coefficients.

Then, the risk difference satisfies

$$\mathcal{R}_K - \mathcal{R}_1 \leq -\Big\|\sum_{k=2}^{K_*} \Theta_k \mathbf{A}^k \mathbf{X}_t\Big\|_{L^2}^2 + c_1\frac{\sigma^2}{T}\sum_{k=2}^{K} \kappa_k(\mathbf{A}, \mathbf{X}_t) - c_2\frac{\sigma^2}{T}\sum_{k=2}^{K} \Delta_k(\mathbf{A}),$$

for constants $c_1, c_2 > 0$ independent of $n, T$.

Here:

- $\Theta_k$ denotes the ground-truth filter coefficient at order $k$;

- $\kappa_k(\mathbf{A}, \mathbf{X}_t)$ quantifies the conditioning of $k$-th order feature covariance;

- $\Delta_k(\mathbf{A}) \lesssim d_{\min}^{-1}\sum_i |\lambda_i(\mathbf{A})|^{2k}$ captures variance suppression due to multi-hop averaging, with $\lambda_i(\mathbf{A})$ the eigenvalues of $\mathbf{A}$.

In particular, for sufficiently large $T$, $K \geq K_*$, and non-trivial minimum degree $d_{\min}$, the right-hand side is negative, implying

$$\mathcal{R}_K < \mathcal{R}_1,$$

i.e., explicitly encoding higher-order adjacency reduces forecasting error under rapidly fluctuating correlation dynamics.

**Theorem 6.** Consider a correlation graph $G$ (without self-loops) whose adjacency is decomposed as $\mathbf{A} = (\mathbf{W}_{\text{diag}}\mathbf{A}_{\text{diag}} + \mathbf{W}_{\text{off}}\mathbf{A}_{\text{off}})$, where $\mathbf{A}_{\text{diag}}$ captures stationary (self/variance) terms and $\mathbf{A}_{\text{off}}$ encodes dynamic cross-variable correlations. Assume time series $\mathbf{x}_t = \mathbf{s}_t + u\,\mathbf{g}_t + \varepsilon_t$, the minimum node degree is $d_{min}$, with metric TCV. Then, as TCV increases, a GCN layer that only embeds $\mathbf{A}_{\text{diag}}\mathbf{X}\mathbf{W}$ is less expressive (i.e., empirical risk defined in [1] if higher than a layer that separates

them, e.g., $\mathbf{W}_{\mathrm{diag}}\mathbf{A}_{\mathrm{diag}}\mathbf{X} + \mathbf{W}_{\mathrm{off}}\mathbf{A}_{\mathrm{off}}\mathbf{X}$, where $+$ denotes addition before applying a non-linear transformation.

*Proof.*
$$\widehat{\mathbf{X}}_{t+1}^{\mathrm{diag}} = \sigma(\mathbf{A}_{\mathrm{diag}}\mathbf{X}_t\mathbf{W}), \qquad \widehat{\mathbf{X}}_{t+1}^{\mathrm{sep}} = \sigma(\mathbf{W}_{\mathrm{diag}}\mathbf{A}_{\mathrm{diag}}\mathbf{X}_t + \mathbf{W}_{\mathrm{off}}\mathbf{A}_{\mathrm{off}}\mathbf{X}_t),$$
$\sigma$ is 1-Lipschitz, $\mathbf{x}_t = \mathbf{s}_t + u\,\mathbf{g}_t + \boldsymbol{\varepsilon}_t$, $\mathrm{Cov}(\boldsymbol{\varepsilon}_t) = \mathbf{D}$ (diagonal), $d_{\min} > 0$.

$$\mathcal{H}_{\mathrm{diag}} \subset \mathcal{H}_{\mathrm{sep}} \quad (\text{set } \mathbf{W}_{\mathrm{off}} = \mathbf{0}) \;\Rightarrow\; \mathrm{Bias}_{\mathrm{sep}}^2 \leq \mathrm{Bias}_{\mathrm{diag}}^2.$$

Let $f^\star(\mathbf{X}_t) = \mathbb{E}[\mathbf{X}_{t+1} \mid \mathbf{X}_t]$. For any graph-level scalar task $y$, with $\Delta = u\,\mathbf{g}_t + \boldsymbol{\varepsilon}_t$, $\Sigma = \mathrm{Cov}(\Delta) = u^2\Sigma_{\mathrm{off}} + \mathbf{D}$,

$$\mathbb{E}[y(\mathbf{X}_t + \Delta) - y(\mathbf{X}_t) \mid \mathbf{X}_t] \approx \tfrac{1}{2}\mathrm{Tr}\big(H(\mathbf{X}_t)\Sigma\big) = \tfrac{1}{2}\left(\sum_u H_{uu}\Sigma_{uu} + \sum_{u \neq v} H_{uv}\Sigma_{uv}\right).$$

Since $\Sigma_{uv} = u^2(\Sigma_{\mathrm{off}})_{uv}$ for $u \neq v$, the mixed Hessian couples to *off-diagonal* covariance. Thus, locally (linearizing $\sigma$),
$$f^\star(\mathbf{X}_t) = f^\star_{\mathrm{diag}}(\mathbf{A}_{\mathrm{diag}}\mathbf{X}_t) + u^2\,f^\star_{\mathrm{off}}(\mathbf{A}_{\mathrm{off}}\mathbf{X}_t) + r_t.$$

Let $\Pi_{\mathrm{diag}}$ be the $L^2$-projection onto the span of $\{\mathbf{A}_{\mathrm{diag}}\mathbf{X}_t\}$. Then
$$\inf_{\mathbf{W}}\mathbb{E}\big\|f^\star(\mathbf{X}_t) - \sigma(\mathbf{A}_{\mathrm{diag}}\mathbf{X}_t\mathbf{W})\big\|_F^2 \;\geq\; \big\|(I - \Pi_{\mathrm{diag}})\big(u^2 f^\star_{\mathrm{off}}(\mathbf{A}_{\mathrm{off}}\mathbf{X}_t)\big)\big\|_{L^2}^2 \;\geq\; c_0\,u^2\,\mathrm{TCV},$$

where TCV is any monotone off-diagonal strength (e.g. $\|\mathbf{A}_{\mathrm{off}}\mathbf{X}_t\|_{L^2}^2$ or $\mathrm{Tr}(H\,\Sigma_{\mathrm{off}})$).

Let $\Phi_{\mathrm{off}} = \mathbf{A}_{\mathrm{off}}\mathbf{X}_t$. Degree averaging gives
$$\mathrm{Var}\big[(\Phi_{\mathrm{off}})_{i\cdot}\big] \;\lesssim\; \frac{\sigma^2}{d_i} \quad\Rightarrow\quad \Big\|\frac{1}{T}\sum_t \Phi_{\mathrm{off}}^\top\Phi_{\mathrm{off}} - \mathbb{E}[\Phi_{\mathrm{off}}^\top\Phi_{\mathrm{off}}]\Big\|_{\mathrm{op}} \;\lesssim\; \frac{\sigma^2}{T\,d_{\min}},$$
hence
$$\mathrm{Var}(\mathcal{H}_{\mathrm{sep}}) - \mathrm{Var}(\mathcal{H}_{\mathrm{diag}}) \;\leq\; c_1\,\frac{\sigma^2}{T\,d_{\min}}.$$

With $\mathcal{R} = \mathrm{Bias}^2 + \mathrm{Var} + \mathrm{noise}$,
$$\mathcal{R}_{\mathrm{sep}} - \mathcal{R}_{\mathrm{diag}} \;\leq\; -c_0\,u^2\,\mathrm{TCV} + c_1\,\frac{\sigma^2}{T\,d_{\min}} \quad\Rightarrow\quad \mathcal{R}_{\mathrm{sep}} < \mathcal{R}_{\mathrm{diag}}$$
if $u^2\,\mathrm{TCV} > \frac{c_1}{c_0}\frac{\sigma^2}{T\,d_{\min}}. \to d_{\min} > \frac{c_0}{c_1}\frac{T\,u^2\,\mathrm{TCV}}{\sigma^2}$ $\qquad\qquad\qquad\qquad\square$

## D EMPIRICAL SETUP

### D.1 EVALUATION ON SYNTHETIC BENCHMARK

**Hyperparameter Tuning.** We only report a subset of the tuned hyperparameters, which includes but not limited to: learning rate $10^{(-2\sim-4)}$, hidden embedding size $2^{8\sim10}$, batch size $2^{4\sim9}$, number of encoder and decoder layers $1 \sim 3$, number of attention heads $2 \sim 8$, kernel sizes $3 \sim 7$ and rolling window sizes $12 \times (1 \sim 3)$. For the electricity, solar and ETT datasets, we report the best results obtained from Yi et al. (2023); Wu et al. (2023; 2020a).

**Details of Synthetic Data Generation** At time step $t$, the signal $X(t) \in \mathbb{R}^{N \times 1}$ is sampled as $X(t) \sim \mathcal{N}\big(W(t-1)X(t-1),\,\sigma\big)$, where $\mathcal{N}$ denotes the normal distribution, $\sigma \in \mathbb{R}$ specifies the variance, and $X(0)$ is drawn from a discrete uniform distribution, $W(t-1)$ denotes a transition matrix indicates the underlying graph topology behind time series. We construct $N_w$ transition graphs $(G^{(1)}, \ldots, G^{(N_w)})$, each encoding the structural and transitional statistics of the series within a specific interval. Random walk sequences are generated on these graphs, forming the temporal dynamics: variables correspond to independent walks but share the same transition graph within an interval. Given a period $T_p \geq N_w$, we divide it into $N_w$ equal subintervals of length $T_s$. Each subinterval is assigned a transition matrix, $W(t) = G\Big(\big\lfloor \frac{t \bmod T_p}{T_s} \big\rfloor\Big)$, with $\lfloor\cdot\rfloor$ denoting the floor

function. Within each $T_s$-interval, we simulate $M$ independent random walks as the $M$-dimensional time series. The parameter $N_w$ directly governs dependency complexity: $N_w = T_p$ reduces to the static case (a single fixed transition graph), while smaller $N_w$ induces faster switching, yielding higher-order correlations. To further increase stochasticity and realism, we incorporate random rewiring and Bernoulli noise. We highlight two extreme regimes of this construction and generate six multivariate time series datasets of length 2400. Each dataset is split into training, validation, and test subsets with a 7:1:2 ratio. Full algorithmic details, we provide algorithm table in Algo. D.1

---

**Algorithm 1** Generating MTS Data with the NPR Model

---

**Require:** Total length $T$ of the MTS data, number of variables $N$, number of constant matrices $N_w$
**Require:** Cycle length $T_p$, standard deviation $\sigma$, matrix sparsity threshold $\delta$
**Ensure:** Synthetic MTS data $\mathbf{X} \in \mathbb{R}^{T \times N}$
 1: Generate random orthogonal matrix $\mathbf{P} \in \mathbb{R}^{N \times N}$
 2: **for** $i = 1$ to $N_w$ **do**
 3:     $\boldsymbol{\Sigma}_i = \mathrm{diag}(|\mathcal{N}(0,1)|, \dots, |\mathcal{N}(0,1)|)$
 4:     $\mathbf{G}_i = \mathbf{P}^\top \boldsymbol{\Sigma}_i \mathbf{P}, \alpha = 0$
 5:     $\mathbf{G}_i[\mathbf{G}_i < \alpha] \leftarrow 0$
 6:     **while** $\mathrm{sparsity}(\mathbf{G}_i) > \delta$ **do**
 7:         $\alpha \leftarrow \alpha + 0.02$
 8:         $\mathbf{G}_i[\mathbf{G}_i < \alpha] \leftarrow 0$
 9:     **end while**
10:     $\mathbf{G}_i \leftarrow$ symmetric normalized Laplacian of $\mathbf{G}_i$
11: **end for**
12: Initialize $\mathbf{X} \leftarrow \mathbf{0} \in \mathbb{R}^{T \times N}$
13: $T_l \leftarrow T/T_p$
14: **for** $t = 1$ to $T$ **do**
15:     **if** $(t-1) \bmod T_p = 0$ **then**
16:         Initialize $\mathbf{x} \in \mathbb{R}^N$ randomly from $\{-1, -0.5, 0.5, 1\}$
17:     **else**
18:         $\mathbf{x} \sim \mathcal{N}\left(\mathbf{G}_{(t-1) \bmod T_p / T_l} \cdot \mathbf{X}[t-1], \sigma\right)$
19:     **end if**
20:     $\mathbf{X}[t] \leftarrow \mathbf{x}$
21: **end for**

---

**Implementation** All models are trained under a standardized setup with temporal splits of 60%/20%/20%, and early stopping of epoch 5 is applied based on validation Mean Absolute Error (MAE). We adopt the same hyperparameter grids for parameters embedding sizes $\{32, 128, 512\}$, hidden sizes $\{16, 64, 128\}$, learning rates $\{10^{-3}, 10^{-4}, 10^{-5}\}$, and batch sizes $\{32, 64, 128, 512\}$.

### D.2 MORE DETAILS OF RQ2

**Datasets & Metrics** We consider dataset *Traffic*, *Solar-Energy*, *Electricity* and *PEMS-BAY*. We use the same data splits (70%/20%/10%) as (Wang et al., 2023) for all models and report MAE and RMSE of forecasting horizons with 12 time steps on five random seeds with enabled early stop.

**Baselines** We compare DYNGCN with the strong baselines in short term forecasting, similar to (Wu et al., 2020a), which including An auto-regressive model, VAR-MLP: A hybrid model of the multilayer perception (Zhang, 2003), A Gaussian Process (Frigola, 2015), LSTNet utilizing recurrent neural networks for cross variable correlation (Lai et al., 2018a), TPA-LSTM: An attention-based recurrent neural network (Shih et al., 2019) and strongest baseline MTGNN (Wu et al., 2020a).

**Hyperparameter Tuning.** We performed a grid search over a wide range of hyperparameter values for all reported models. Due to limited space, we only report a subset of the tuned hyperparameters, which includes but not limited to: learning rate $10^{(-2 \sim -4)}$, hidden embedding size $2^{8 \sim 10}$, batch size $2^{4 \sim 9}$, number of encoder and decoder layers $1 \sim 3$, number of attention heads $2 \sim 8$, kernel sizes $3 \sim 7$ and rolling window sizes $12 \times (1 \sim 3)$. For the baseline result, we report the best results obtained from Yi et al. (2023); Wu et al. (2023; 2020a). We provide the details of data preprocessing, split and hyperparameter tuninig in App. D.3

### D.3 More Details of RQ3

**Dataset & Metrics.** We finally evaluate DYNGCN on a real-world high-TCV data set. Specifically, we select a subset from energy datasets in (Shao, 2025), *Germany*, *France* and two public datasets (electricity and solar). It records electricity production in Germany and France in all generation types for all available years. We use the consistent data splits (70%/20%/10%) for all models and report *Mean Absolute Error (MAE)* and *Root Mean Squared Error (RMSE)* of forecasting horizons with 12 time steps on five random seeds with enabled early stop. The datasets span a wide correlation spectrum. For instance, Germany and France exhibit larger magnitudes of TCV (near 1) compared to the classic datasets (around 0.5), indicating stronger temporal fluctuations and heterogeneous regional dynamics. For results on horizon 3,6 please refer to App. D.3.

**Hyperparameter Tuning.** We performed a grid search over a wide range of hyperparameter values for all reported models. We report the tuned hyper-parameters, which includes but not limited to: learning rate $10^{(-2 \sim -4)}$, hidden embedding size $2^{8 \sim 10}$, batch size $2^{4 \sim 9}$, number of encoder and decoder layers $1 \sim 3$, number of attention heads $2 \sim 8$, kernel sizes $3 \sim 7$ and rolling window sizes $12 \times (1 \sim 3)$. For the electricity, solar and ETT datasets, we report the best results obtained from Yi et al. (2023); Wu et al. (2023).

**Baselines.** We evaluate 20 baselines reported in Tab. 8, which includes three classical statistical methods Sims (1980); Box & Jenkins (1970); Box et al. (2015), three MLP-based approaches Zeng et al. (2023); Oreshkin et al. (2020); Wang et al. (2024) and several RNN/CNN-based architectures Sen et al. (2019); Lai et al. (2018b); Bai et al. (2018); Zhang et al. (2017). We also include three advanced paradigms: 1) Transformer-based models Kitaev et al. (2020); Zhou (2021); Wu et al. (2021); Zhou et al. (2022), which optimize temporal modeling via optimized attention mechanisms; 2) Spectral GNN models Cao et al. (2020); **?**); Yi et al. (2023), which leverage nonlinear filtering in the spectral domain to capture spatial correlations; 3) Spatial graph-based approaches Wu et al. (2019b; 2020b); Liu et al. (2022b), which rely on localized message passing mechanisms to dynamically model and adapt to spatial dependencies.

Table 7: Dataset statistics and corresponding TCV values.

| Dataset | $N$ | Rows | Windows | $B$ | Step | TCV |
|---|---|---|---|---|---|---|
| Germany | 16 | 333,060 | 333,037 | 24 | 1 | **0.997** |
| France | 10 | 83,265 | 83,242 | 24 | 1 | **0.969** |
| exchange_rate | 8 | 7,588 | 7,565 | 24 | 1 | **0.007** |
| electricity | 321 | 26,304 | 26,281 | 24 | 1 | **0.194** |
| traffic | 862 | 17,544 | 1,241 | 24 | 1 | **0.679** |
| solar | 137 | 52,560 | 12,342 | 24 | 1 | **0.323** |

Table 8: Mean MAE/RMSE for two dynamic-correlation energy datasets. We report MAE, RMSE and MAPE at horizons 6 and 12. Best, second-best and third-best results are highlighted.

| Methods | Germany (TCV = 1.097) | | | | | | France (TCV = 0.970) | | | | | | Rank |
| | H=6 | | | H=12 | | | H=6 | | | H=12 | | | |
| | MAE | RMSE | MAPE | MAE | RMSE | MAPE | MAE | RMSE | MAPE | MAE | RMSE | MAPE | |
|---|---|---|---|---|---|---|---|---|---|---|---|---|---|
| VAR | 0.210 | 0.340 | 0.191 | 0.243 | 0.381 | 0.215 | 0.155 | 0.241 | 0.132 | 0.177 | 0.260 | 0.151 | 11 |
| DLinear | 0.230 | 0.320 | 0.201 | 0.266 | 0.368 | 0.231 | 0.170 | 0.230 | 0.140 | 0.196 | 0.259 | 0.158 | 12 |
| LSTNet | 0.165 | 0.290 | 0.145 | 0.193 | 0.346 | 0.172 | 0.150 | 0.220 | 0.129 | 0.177 | 0.263 | 0.146 | 9 |
| Autoformer | 0.177 | 0.325 | 0.157 | 0.204 | 0.376 | 0.181 | 0.143 | 0.230 | 0.122 | 0.165 | 0.263 | 0.141 | 8.5 |
| Informer | 0.248 | 0.299 | 0.211 | 0.283 | 0.324 | 0.239 | 0.119 | 0.194 | 0.102 | 0.137 | 0.217 | 0.116 | 5 |
| Reformer | 0.259 | 0.331 | 0.222 | 0.297 | 0.361 | 0.247 | 0.124 | 0.208 | 0.106 | 0.141 | 0.233 | 0.119 | 10 |
| FourierGNN | 0.097 | 0.166 | 0.086 | 0.110 | 0.186 | 0.098 | 0.084 | 0.143 | 0.073 | 0.096 | 0.164 | 0.083 | 3 |
| StemGNN | 0.155 | 0.259 | 0.137 | 0.179 | 0.285 | 0.158 | 0.128 | 0.189 | 0.109 | 0.148 | 0.206 | 0.124 | 7.5 |
| TPGNN | 0.086 | 0.152 | 0.079 | 0.099 | 0.173 | 0.090 | 0.077 | 0.139 | 0.070 | 0.089 | 0.158 | 0.081 | 2 |
| MTGNN | 0.013 | 0.031 | 0.030 | 0.016 | 0.034 | 0.038 | 0.010 | 0.020 | 0.070 | 0.012 | 0.023 | 0.093 | 2.5 |
| GraphWaveNet | 0.009 | 0.021 | 0.028 | 0.013 | 0.028 | 0.040 | 0.010 | 0.020 | 0.069 | 0.012 | 0.025 | 0.092 | 1.5 |
| DYNGCN +1 | 0.009 | 0.021 | 0.028 | 0.012 | 0.027 | 0.038 | 0.009 | 0.019 | 0.068 | 0.012 | 0.023 | 0.084 | 1 |
| DYNGCN +2 | 0.008 | 0.019 | 0.023 | 0.010 | 0.022 | 0.030 | 0.008 | 0.016 | 0.060 | 0.010 | 0.020 | 0.079 | 0.5 |

## E   ABLATION STUDY

We evaluate the incremental contributions of our two design choices under the same training protocol. Recall that **D1** removes the second (adaptive/temporal) graph and keeps a single fixed/base graph; **D2** denotes dual-graph variants with targeted edits: (*i*) **D2-rand** replaces the base graph with a random topology while retaining the adaptive branch, and (*ii*) **D2-noTemp** disables the adaptive/temporal branch while keeping the base graph. Unless otherwise noted, depth and optimization are

Table 9: REAL DATA: EXCHANGE RATE — Prefix-mean MAE/RMSE over the first $H$ horizons.

| Dataset | | Exchange Rate | | | |
|---|---|---|---|---|---|
| Methods | Metrics | H=1 | H=3 | H=6 | H=12 |
| D2 w. random | MAE | 0.0070 | 0.0098 | 0.0093 | 0.0109 |
| | RMSE | 0.0090 | 0.0128 | 0.0126 | 0.0151 |
| D2 w. graph | MAE | 0.0094 | 0.0091 | 0.0114 | 0.0117 |
| | RMSE | 0.0127 | 0.0141 | 0.0156 | 0.0153 |
| D2 Ours | MAE | 0.0062 | 0.0079 | 0.0095 | 0.0088 |
| | RMSE | 0.0088 | 0.0104 | 0.0133 | 0.0126 |

identical across variants. For tables reporting $H \in \{1, 3, 6, 12\}$, the value at $H=K$ is the *prefix-mean* over the first $K$ horizons.

**Results on Synthetic.**   Across difficulty levels, the full model consistently yields the lowest MAE/RMSE at both short and long horizons. On the Easy split, our model is best at $H=1$ and maintains the lead through $H=12$; as difficulty increases (Medium $\rightarrow$ Hard/Very Hard), the margin becomes more visible. These trends support that (i) keeping two graphs and (ii) mixing multi-hop information are complementary; removing either part (**D1** or **D2-noTemp**) hurts most at short horizons, while corrupting the base topology (**D2-rand**) mainly degrades late-horizon stability.

**Results on Exchange Rate.**   We observe the same pattern. The full model (SERIESGCN) outperforms **D1** and **D2** for $H=1, 3, 6, 12$. **D2-noTemp** is notably worse at $H=1 \sim 3$, indicating the adaptive/temporal branch is crucial for near-term predictions. **D2-rand** remains competitive at short horizons (the adaptive branch compensates for an imperfect base graph) but lags behind at $H=12$. Overall, dual-graph aggregation with power mixing is the most robust across horizons.

**Depth & order.**   Reducing each block from two spatial layers to one uniformly increases error—especially at $H=12$—demonstrating that per-block spatial depth stabilizes multi-step rollout. Varying the spatial order (e.g., $o=1$ vs. $o=2$) yields consistent but smaller gains than keeping two graphs: moderate order together with dual graphs offers a favorable accuracy–efficiency trade-off.

Table 10: SYNTHETIC: Ablation on graph variants under varying TGV.

| Dataset | | Easy | | | | Medium | | | | Hard | | | | Very Hard | | | |
|---|---|---|---|---|---|---|---|---|---|---|---|---|---|---|---|---|---|
| Methods | Metrics | 1 | 3 | 6 | 12 | 1 | 3 | 6 | 12 | 1 | 3 | 6 | 12 | 1 | 3 | 6 | 12 |
| D2 Ours w. random graph | MAE | 0.192 | 0.248 | 0.298 | 0.351 | 0.291 | 0.370 | 0.427 | 0.490 | 0.336 | 0.416 | 0.463 | 0.498 | 0.410 | 0.464 | 0.482 | 0.493 |
| | RMSE | 0.384 | 0.425 | 0.460 | 0.496 | 0.489 | 0.539 | 0.575 | 0.600 | 0.516 | 0.568 | 0.574 | 0.605 | 0.551 | 0.566 | 0.573 | 0.580 |
| D2 Ours w.o. remove graph second graph | MAE | 0.193 | 0.249 | 0.300 | 0.352 | 0.292 | 0.369 | 0.429 | 0.489 | 0.338 | 0.414 | 0.467 | 0.503 | 0.414 | 0.465 | 0.486 | 0.497 |
| | RMSE | 0.386 | 0.427 | 0.464 | 0.492 | 0.494 | 0.542 | 0.578 | 0.606 | 0.521 | 0.569 | 0.577 | 0.606 | 0.553 | 0.570 | 0.577 | 0.585 |
| D1 Ours w.o 2nd graph | MAE | 0.190 | 0.246 | 0.297 | 0.350 | 0.289 | 0.369 | 0.425 | 0.488 | 0.335 | 0.415 | 0.461 | 0.496 | 0.408 | 0.462 | 0.480 | 0.491 |
| | RMSE | 0.382 | 0.423 | 0.458 | 0.494 | 0.488 | 0.537 | 0.573 | 0.598 | 0.514 | 0.566 | 0.572 | 0.602 | 0.549 | 0.564 | 0.571 | 0.578 |

Table 11: SYNTHETIC: Ablation on simplified architectures under varying TGV.

| Dataset | | Easy | | | | Medium | | | | Hard | | | | Very Hard | | | |
|---|---|---|---|---|---|---|---|---|---|---|---|---|---|---|---|---|---|
| Methods | Metrics | 1 | 3 | 6 | 12 | 1 | 3 | 6 | 12 | 1 | 3 | 6 | 12 | 1 | 3 | 6 | 12 |
| only 1 GCN Layer | MAE | 0.193 | 0.249 | 0.300 | 0.352 | 0.292 | 0.369 | 0.429 | 0.489 | 0.338 | 0.414 | 0.467 | 0.503 | 0.414 | 0.465 | 0.486 | 0.497 |
| | RMSE | 0.386 | 0.427 | 0.464 | 0.492 | 0.494 | 0.542 | 0.578 | 0.606 | 0.521 | 0.569 | 0.577 | 0.606 | 0.553 | 0.570 | 0.577 | 0.585 |
| only 1rd Adj | MAE | 0.190 | 0.246 | 0.297 | 0.350 | 0.289 | 0.369 | 0.425 | 0.488 | 0.335 | 0.415 | 0.461 | 0.496 | 0.408 | 0.462 | 0.480 | 0.491 |
| | RMSE | 0.382 | 0.423 | 0.458 | 0.494 | 0.488 | 0.537 | 0.573 | 0.598 | 0.514 | 0.566 | 0.572 | 0.602 | 0.549 | 0.564 | 0.571 | 0.578 |
| Ours | MAE | 0.192 | 0.248 | 0.298 | 0.351 | 0.291 | 0.370 | 0.427 | 0.490 | 0.336 | 0.416 | 0.463 | 0.498 | 0.410 | 0.464 | 0.482 | 0.493 |
| | RMSE | 0.384 | 0.425 | 0.460 | 0.496 | 0.489 | 0.539 | 0.575 | 0.600 | 0.516 | 0.568 | 0.574 | 0.605 | 0.551 | 0.566 | 0.573 | 0.580 |