# OpenReview forum: "DynGCN: Capturing Dynamic Correlation with Message Passing"
_ICLR.cc/2026/Conference — ICLR 2026 Conference Withdrawn Submission_

### Official Review · Reviewer_nhq3 · 2025-10-29

**Soundness:** 3
**Presentation:** 3
**Contribution:** 2
**Rating:** 4
**Confidence:** 4

**Summary:**

The authors focus on graph-based time series forecasting, in the absence of underlying graph structure, and propose a GNN-based framework that captures static and dynamic time series correlations over time, beyond static graph topology. With extensive theoretical formulations, the authors propose an architectural design that incorporates high-order correlation-based message passing and considers static and dynamic propagation separation in the message passing. These considerations are shown to improve forecasting in the presence of dynamic correlations, in synthetic and real-world experiments against baselines, with significant margins in some cases.

**Strengths:**

- **[S1: Quality - Mathematical Framework]** Different aspects of the proposed design choices are mathematically proven and connected to common time series statistical properties rather than heuristically chosen.
- **[S2: Significance - Performance Improvements]** The proposed method achieves significantly improved performance against several baselines for several datasets.
- **[S3: Clarity]** The paper is easy to follow, and the different points made are well-justified via experiments or mathematical proofs.

**Weaknesses:**

- **[W1: Clarity - Positioning in Graph Structure Learning]** The positioning of the authors against related works in graph structure learning for time series forecasting is rather poor, both in the introduction and the related works section. Specifically, the authors mention that related works build on static graphs, disregarding dynamic correlations, yet those are assumed to be extracted via the raw time series sliding windows used as dynamic node features. Additionally, no distinction on the type of the structure extracted, fixed (based on time series correlation, such as FourierGNN) vs learnable (top-K operation on the transformed input vectors, such as MTGNN or learned with gumbel softmax, such as GTS (Shang et al, 2021)) is made with respect to how this affects correlation learning. This gap should be explicitly tackled in the relevant sections (e..g, Graph Topology Identification) to enable proper positioning of the paper.
- **[W2: Clarity & Quality - Fully-Connected Adjacency Structure]** It seems that the authors focus on fully-connected weight adjacencies, specifically $X^TX$. Dense graph structures have been criticized in the relevant literature as backpropagating spurious relationships in time series forecasting, while hindering the interpretability aspect of the learned correlations, which is the main goal of non-structured time series graph-based forecasting and which can indeed be enabled via sparse learnable graphs (Cini et al., 2023). Addressing this design choice is crucial for the motivation behind the paper and its positioning in the time series graph-based literature. It is also unclear how the architectural designs can scale for large time series lengths and a high number of variables.
- **[W3: Quality & Significance - Adjacency Separation]** The disentanglement in the message passing step for static and dynamic correlations is indeed interesting, while being also one of the main contributions of the paper, yet its effect seems incremental in the ablation study of Table 4. At the same time, this separation doubles the parameters of the message passing layer of the proposed model. Since no normalization is done, it is unclear if one of the parts can dominate the sum at the end.
- **[W4: Quality & Significance - Time Cost Comparisons]** The authors should consider including time cost/memory comparisons with relevant baselines and the datasets used, to showcase how their architectural choices affect scalability in practice, combined with the achieved performance.
- **[W5: Quality & Significance - Experimental Evaluation]** Some considered baselines in the time series forecasting literature are slightly outdated when it comes to sequential architectures (see more recent SOTA here: https://github.com/thuml/Time-Series-Library). Several benchmark datasets are also not considered (see datasets in FourierGNN and standard datasets in the time series community, such as ETT, Weather). Additional GNN-based baselines are also usually considered (Bai et al., 2020; Wu et al., 2019).
- **[W6: Significance - Results]** The authors should show standard deviations for some datasets where performance improvements compared to baselines are small (at least in the appendix).

--Shang, C., Chen, J., & Bi, J. (2021). Discrete graph structure learning for forecasting multiple time series. arXiv preprint arXiv:2101.06861.
Cini, A., Zambon, D., & Alippi, C. (2023). Sparse graph learning from spatiotemporal time series. Journal of Machine Learning Research, 24(242), 1-36.

--Bai, L., Yao, L., Li, C., Wang, X., & Wang, C. (2020). Adaptive graph convolutional recurrent network for traffic forecasting. Advances in neural information processing systems, 33, 17804-17815.

--Wu, Z., Pan, S., Long, G., Jiang, J., & Zhang, C. (2019). Graph wavenet for deep spatial-temporal graph modeling. arXiv preprint arXiv:1906.00121.

**Questions:**

1. **[Based on W2]** Without any regularization/sparsification on the adjacency, it is unclear how the method differs from sequential models, eg, transformers, in capturing correlations for time series forecasting.
2. **[Based on W1/W2]** Can the authors show visualizations of the correlations learned in forecasting? Could those be explainable when it comes to the input data and the task at hand?
3. **[Based on W2/W4]** In the ablation, there is a small drop in performance when the two main architectural design choices are removed from the proposed method. Could the authors justify better where the large performance improvements for some datasets mainly come from?

---

### Official Review · Reviewer_mXrX · 2025-10-30

**Soundness:** 1
**Presentation:** 1
**Contribution:** 1
**Rating:** 0
**Confidence:** 4

**Summary:**

The paper addresses the problem of modeling dynamic dependencies in multivariate time series by relying on graph representations. The proposed method models aims at modeling both dynamic and static edges within a message-passing framework. However, as discussed below, the paper suffers from severe issues in presentation, conceptualization, and experimental design.

**Strengths:**

* Accounting for dynamic inter-series relationships in time series forecasting is an important and relevant problem.

**Weaknesses:**

One of the main issues of the paper is that most of the results concern modeling covariance, yet the paper presents them from the beginning as modeling generic relationships among time series. This leads to several conceptual problems.

* **Poor presentation leading to several conceptual issues.** The structure and presentation of the paper are confusing and lack clarity in several respects.
    - The paper seems to use multiple, incompatible definitions of what an adjacency matrix is. In Section 2, the paper “defines” the adjacency matrix as the sample covariance. This is a major conceptual limitation, as covariance only accounts for specific linear dependencies, whereas the graph representations typically used in this context often model more complex relationships (e.g., the structure of a traffic network or spatial proximity in spatiotemporal data). Indeed, in lines 96–100, the paper states that existing methods use adjacency matrices that are not compatible with the definition in Eq. 1. This creates significant ambiguity throughout the paper.  For instance, Section 2.1 appears to list properties of the covariance matrix under certain assumptions on the data-generating process. Overall, conflating the dependencies modeled by a graph structure with covariance estimation is a major limitation, and the paper is ambiguous on this point. The presentation would be much clearer if the paper referred directly to the problem of estimating covariance matrices rather than generic graph structures and used the term “covariance matrix” instead of “adjacency matrix.” This would also allow for a more appropriate contextualization relative to related work.
    - In Section 2.2, the score in Eq. (2) can again only model linear dependencies — a major limitation that should be acknowledged and discussed.
    - It is unclear how the static and dynamic adjacency matrices are obtained. There is no precise mathematical definition of either anywhere in the paper.
    - It is also unclear how some of the theorems relate to the problems the paper claims to address, and several results are trivial given the assumptions made. For example, Theorem 1 states that if we assume the data are generated by a polynomial graph filter, then using a polynomial graph filter as a model is better than using a first-order convolution. This is a tautology. Similarly, assuming that the true adjacency can be decomposed into diagonal and off-diagonal terms, Theorem 2 concludes that using both is better than using only the diagonal terms. Again, this is trivial.

* **Poor empirical design.** The empirical evaluation has several serious issues.
    - No standard deviations are reported for any of the results, making it difficult to draw meaningful conclusions. Moreover, the paper states that it directly uses results from prior works such as FourierGNN and StemGNN. This is problematic for several reasons: (1) even small differences in preprocessing, batching, or data splitting can lead to significantly different results, making a direct comparison meaningless; and (2) both FourierGNN and StemGNN suffer from reproducibility problems, with many open GitHub issues reported in their repositories. For instance, a known bug in StemGNN allowed the model to access future data. This makes the reported empirical results difficult to trust.
    - Showing that model performance deteriorates when the correlation structure becomes more dynamic (Tables 1 and 2) is unsurprising, as the task itself becomes harder. Indeed, performance deteriorates for all models, including DynGCN, and the largest performance gap in Table 2 w.r.t. TPGNN occurs for the lowest level of “TCV”.

* **Limited technical contribution.** The use of polynomial graph filters is standard in many existing architectures, as acknowledged in the paper. Combining this with the use of different adjacency matrices is not particularly novel. Theorem 2 seems to imply that the static adjacency matrix is diagonal, which would make it equivalent (up to rescaling) to the identity matrix — effectively implementing a skip connection.

These flaws make the paper fall well below the acceptance threshold.

Additional comments:

- Broken reference in line 219.

**Questions:**

See weaknesses.

---

### Official Review · Reviewer_ka7G · 2025-10-31

**Soundness:** 3
**Presentation:** 3
**Contribution:** 3
**Rating:** 6
**Confidence:** 4

**Summary:**

The paper proposes DYNGCN, a graph neural network (GNN) framework for multivariate time series forecasting (MTSF) that explicitly models dynamic correlations between variables. Traditional GNN-based MTSF methods assume a static correlation structure, limiting their effectiveness when relationships evolve over time. The framework introduces two key components: High-order correlation–based message passing and Static and dynamic propagation separation. The paper also introduces Temporal Correlation Volatility (TCV) to quantify how much pairwise correlations change over time, linking it to model performance degradation. The theoretical proofs and empirical evaluation on synthetic and six real-world datasets show significant performance gains over 21 baselines, including GNNs and Transformers.

**Strengths:**

1.	The paper addresses a key limitation of existing GNN-based forecasters: the assumption of static correlations. It convincingly argues and demonstrates that dynamic correlations degrade performance of many state-of-the-art GNN and Transformer baselines (e.g., Informer, Autoformer, MTGNN).

2.	Strong clarity in theoretical guarantees on the properties of the time-varying adjacency matrix (Locality, Soft-Transitivity, Diagonal-Dominance, Additivity).
3.	The paper presents a novel quantitative measure (TCV), a model-agnostic measure to capture dynamic correlation shifts. It seems to add diagnostic value to time series analysis.
4.	Experiments are extensive and include memory and computational analysis.

**Weaknesses:**

1.	Appendix is cited in the paper, but it is not attached. I see some major flaws because of this. It makes it impossible to verify proofs, hyperparameter settings, and some theoretical derivations.

2.	This omission significantly affects reproducibility and completeness.

3.	Technical details are not clear/ incomplete. So, reproducibility is a concern. For e.g., the construction of the dynamic adj_matrix is not fully described. Does it rely on sliding-window covariance, online learned embeddings, or temporal smoothing? How are the polynomial coefficients are not described too.

4.	Covariance vs. Correlation: Definition 1 defines the adjacency matrix as the sample covariance (A = (1/T)XX^T), but the paper repeatedly refers to "correlation." Covariance and correlation are different. And the mixed usage is confusing here. Or will both produce similar results? (Normalized vs unnormalized?)

5.	Some reported relative improvements (e.g., 98% MAE reduction on the Electricity dataset) seem implausibly large compared to prior literature. 98% improvement sounds unreasonable, or it is a groundbreaking methodology (not supported by other results)

6.	The robustness analysis to noise and missing data is important for validating dynamic scenarios as mentioned in the problem statement. Also, the stability/scalability analysis.

7.	In page 4, Theorem 1: The theorem claims that high-order GCN layers reduce prediction error under certain conditions, but the proof is missing or not cited correctly.

8.	In page 5, theorem 2 proof is missing. Table 9 is cited but not present in the paper. The connection between "expressiveness" and "empirical risk" is unclear, and the assumption of a linear signal model undermines generalizability.

9.	No visualization of correlation graph dynamics or qualitative examples of how TCV affects learned structure. Adding heatmaps or temporal edge-weight visualizations would strengthen insight.

**Questions:**

Refer the weaknesses.

---

### Official Review · Reviewer_o9FS · 2025-11-03

**Soundness:** 1
**Presentation:** 2
**Contribution:** 2
**Rating:** 2
**Confidence:** 4

**Summary:**

This paper studies the problem of modeling dynamic correlations in multivariate time series forecasting. The authors propose DynGCN, a graph neural network framework that captures both high-order and time-varying correlations through two key designs: high-order message passing and static–dynamic propagation separation. The authors also introduce the Temporal Correlation Volatility (TCV) to quantify the degree of correlation dynamics.

**Strengths:**

1. The paper clearly identifies an important issue in multivariate time series forecasting—the dynamic nature of inter-variable correlations—and provides empirical evidence that many existing graph-based models fail under high TCV.

2. The authors provide theoretical analyses supporting the benefits of high-order message passing and static–dynamic propagation separation.

**Weaknesses:**

1. The presentation should be improved. For example, in Section 2, the proposed TCV metric is discussed together with general theoretical foundations, making it difficult to discern what is newly introduced in this work.

2. The paper does not provide a clear definition or construction procedure for the two adjacency components $A_{stat}$ and $A_{dyn}$ in Eq. (7). It is unclear whether they are computed directly from time-varying correlations or learned implicitly through network parameters.

3. The implementation of the DCRNN component appears inconsistent with the description in the paper, as the model files do not contain the core model logic. Moreover, the paper does not clearly explain how DynGCN handles sequential data —treating the graph propagation output directly as the final prediction seems conceptually incomplete or inadequate for a time series forecasting task.

4. At line 300, the paper defines the output as $\hat{Y} = softmax(H^{final} W_c)$ , implying that each node produces a prediction sequence of length $T$, followed by a softmax operation. This design choice is unusual, as softmax is typically unnecessary—and indeed inappropriate—for time series forecasting, where the outputs are continuous values rather than categorical probabilities. Applying softmax forces each node’s predictions to sum to one across the time dimension $\sum_{t=1}^T\hat{Y}_{n,t} = 1$, which treats the output as a probability distribution rather than a regression sequence.

5. The experimental setup does not clearly explain how the sequential data are handled. While the baselines are all time-series forecasting models, the paper does not specify the input and output sequence lengths. In addition, most of the selected baseline methods are outdated.

**Questions:**

See W1-W5.

---

### Note · Authors · 2025-12-01

I have read and agree with the venue's withdrawal policy on behalf of myself and my co-authors.